# Mesoscale spatio-temporal variability of airborne lidar-derived aerosol properties in the Barbados region during EUREC⁴A

5  Patrick Chazette[1], Alexandre Baron[1*], Cyrille Flamant[2]

[1] LSCE/IPSL, CNRS-CEA-UVSQ, University Paris-Saclay, CEA Saclay, 91191 Gif sur Yvette, France

[2] LATMOS/IPSL, CNRS-SU-UVSQ, Sorbonne Université, Campus Pierre et Marie Curie, 10  75252 Paris, France

[*] Now at Laboratoire de l'Atmosphère et des Cyclones, UMR8105, Université de la Réunion - CNRS - Météo-France, Saint-Denis de La Réunion, France

Correspondence to: Patrick Chazette (patrick.chazette@lsce.ipsl.fr)

**Abstract.** From 23 January to 13 February 2020, twenty ATR-42 scientific flights were conducted in the framework of the EUREC4A field campaign over the tropical Atlantic, off the coast of Barbados (-58°30' W 13°30'N). By means of a sideway-pointing lidar, these flights allowed to retrieve the optical properties of the aerosols found in the sub-cloud layer and below 20  the trade winds inversion. Two distinct periods with significant aerosol contents were identified in relationship with the so-called trade wind and tropical regimes, respectively.  For these two regimes, mixings of two air mass types encompassing dust and carbonaceous aerosols have been highlighted. Both were mainly from the West Africa with similar optical contributions and linked to dust uptake above Sahara and biomass burning between Guinea Bissau and Ivory 25  Coast. In the tropical transport regime, the wind within the planetary boundary layer is stronger and favours a contribution of marine aerosols (sulphate and sea salt aerosol components) in shallower aerosol layers than for the trade wind transport regime. The latter is responsible for advecting dust-biomass burning aerosol mixtures in deeper, well-mixed layer, in part due to the complex interactions of the easterly flow from West Africa with mid-latitude dynamics. The 30  aerosol vertical structures appear to be well reproduced using atmospheric composition reanalyses from CAMS when comparing with lidar-derived vertical profiles. The competition between the two types of transport regimes leads to strong heterogeneity in the optical properties of the horizontal aerosol field. Our study highlights the transport regime under which a significant mixture of dust and biomass burning aerosols from West Africa can be observed 35  over the Caribbean, and Barbados in particular, namely the trade wind regime.

Keywords: Trade wind, weather regime, transport, dust, biomass burning, tropical, Atlantic Ocean

## 1   Introduction

In the Caribbean region, mixtures of different types of aerosols have often been observed, especially mixtures of Saharan dust (SD) and biomass burning (BB) aerosols which add to the local oceanic sources of sea salts and sulphates. Long-range transport of SD and BBA aerosols from West Africa across the equatorial North Atlantic occurs all year long but exhibits a marked seasonal cycle. For instance, summertime and wintertime SD aerosol transport characteristics have been shown to differ significantly, with SD being transported at higher latitude and coarser particles being advected further west during the summer (e.g. Van Der Does et al., 2016) in the African Easterly jet-driven Saharan air layer (e.g. Prospero and Carlson, 1972). In contrast, during wintertime, SD is transported at lower altitudes (below 3 km a.m.s.l.) and further south (owing to the equatorward migration of the intertropical Convergence Zone) towards northeast South America (e.g. Ansmann et al., 2009; Baars et al., 2011; SWAP et al., 1992) and the Caribbean (Haarig et al., 2017, 2019). SD in the Caribbean is generally observed to be mixed with BB aerosols from West Africa and South America, with BB-SD mixtures generally being carried above dust layers in the winter (Haarig et al., 2017, 2019; Tesche et al., 2009, 2011; Weinzierl et al., 2017).

One of the first characterization of the SD transport across the tropical Atlantic Ocean was performed by Prospero (1968). Knowing that Africa is the world's largest dust source (Huneeus et al., 2012), this work was then continued by various authors given the climatic and physicochemical significant impacts of these aerosols. SD aerosols play a major role in the primary productivity of the ocean through the enrichment of surface waters by mineral deposits that promote the development of phytoplankton (e.g. Okin et al., 2011). They also contribute to the fertilization of continental surfaces (e.g. Muhs et al., 2007). SD dusts are also the major component of atmospheric aerosol composition over the tropical Atlantic and Caribbean where they play a leading role on the radiative balance (e.g. Li et al., 1996; Prenni et al., 2009) and cyclogenesis (e.g. Zipser et al., 2009). The impact of SD aerosols varies greatly over time over the Caribbean Sea. Indeed, Prospero et al. (2014) showed a dominance of SD from January to May over French Guiana, whereas it is more prevalent from May to September over Martinique. Ben-Ami et al. (2010) identified the Bodélé depression in northern Chad as one of the main sources of SD aerosols also previously identified as such by Koren et al. (2006). Nevertheless,

this source is more likely as the season shifts to summer in Africa. Indeed, sources further north (e.g., in northern Chad, Mali, Mauritania, and southern Algeria) become more active, linked to the development and movement of African easterly waves in concert with extratropical disturbances (Cuesta et al., 2020; Knippertz and Todd, 2010). However, SD aerosols have been shown not to necessarily dominate the atmosphere composition over the tropical Atlantic. As a result, it has recently been shown that BB aerosols have been underestimated as a phosphorus input to surface waters in the tropical Atlantic (Barkley et al., 2019). In fact, a large fraction of the African emissions of SD and BB aerosols are carried across the west coast of North Africa to the western Atlantic. As described by Adams et al. (2012) using CALIOP data, these long-ranged transported aerosols are originating from very intense Saharan desert sources and even from nearby BB areas.

Very quickly, the scientific community became aware of the importance of knowing the vertical distribution of aerosols in order to assess their impact more accurately during their transport across the Atlantic Ocean. The first measurement was made off the West African coast, from the Cape Verde archipelago, where observations were made from a balloon sounding (Dulac et al., 2001), which highlighted a low altitude transport of dust in the trade winds over the tropical Atlantic into the so-called Saharan air layer as initially identified by Carlson and Prospero (1972) during summer. These measurements were very soon followed by numerous and lidar observations across the North Atlantic acquired as part of dedicated campaigns such as SAMUM-2 (Saharan Mineral Dust Experiment (Ansmann et al., 2011)), SALTRACE (Saharan Aerosol Long-Range Transport and Aerosol–Cloud-Interaction Experiment (Weinzierl et al., 2017)) and NARVAL (Next-generation Aircraft Remote-Sensing for Validation Studies (Stevens et al., 2019)). Such observations were made from ground-based lidar measurements in the Cape Verde region (Ansmann et al., 2009, 2011), in Barbados (Groß et al., 2015; Haarig et al., 2017, 2019) and over Amazonia (e.g. Ansmann et al., 2009; Baars et al., 2011), from ship-borne lidar measurements (Rittmeister et al., 2017) or even and from nadir-pointing airborne lidar measurements (e.g; Chazette et al., 2001; Gutleben et al., 2019; Tanré et al., 2003; Weinzierl et al., 2017).

The study presented in this article is a follow-up to the EUREC[4]A (Elucidating the role of clouds-circulation coupling in climate) field campaign (Bony et al., 2017) which took place in January-February 2020 over the western tropical Atlantic, West of Barbados (Stevens et al., 2021). During the airborne measurements performed from the French research aircraft ATR-42 operated by the Service des Avions Français Instrumentés pour la Recherche en Environnement

(SAFIRE), a variety of aerosol optical property signatures were observed (Chazette et al., 2020) and the objective of this paper is to describe their origin and the meteorological conditions that led to their observation over the Barbados region. Airborne observations of atmospheric aerosols off the coast of Barbados are scarce, especially in winter. They are nevertheless relevant to complete the existing body of literature, in particular concerning the satellite observations that are often disturbed by the ubiquitous presence of clouds over this region.

The Section 2 is dedicated to a brief presentation of the lidar used and the flight plans of the SAFIRE ATR-42. The different optical parameters found with the lidar measurements will also be explained. The optical properties of aerosols encountered during the scientific flights will be presented in Section 3. The spatiotemporal evolution of the aerosol layers during transport, prior to their arrival off Barbados, will be presented in Section 4 based on spatial observations and modelling. Section 5 will discuss the results taking into account transport and weather conditions. The conclusion will be presented in Section 6.

## 2    Lidar observations

### 2.1    Instrument

During all the EUREC[4]A field campaign, the Airborne Lidar for Atmospheric Studies (ALiAS Chazette et al. (2012)) was installed in the ATR-42 aircraft of SAFIRE which performed a series of 20 flights off the east coast of Barbados. The lidar system main characteristics and implementation in the aircraft are presented in Chazette et al. (2020). The measurements are performed at 355 nm using an horizontal line-of-slight and they provide access to both the aerosol extinction coefficient and to the linear depolarization of atmospheric aerosols.

### 2.2    Lidar derived aerosol optical properties

The aerosol extinction coefficient (AEC) and volume depolarization ratio (VDR) can be directly deduced from the ALiAS horizontal measurements without any a priori assumptions on the nature of the aerosols (see Chazette et al. (2020) for EUREC[4]A related retrievals). After calibration of the lidar and thus determination of its system constant, the lidar ratio (LR) and particle depolarization ratio (PDR) evaluations can be performed.

### 2.2.1    Direct assessment from each horizontal line-of-sight

The determination of the AEC from horizontal line-of-slight has already been described in Chazette et al. (2007). The calculation is performed by linear fitting on the logarithm of the apparent backscatter coefficient (ApBC), here in the range from 0.2 to 1 km away from the aircraft where the effect of the overlap function of the lidar is negligeable (Chazette et al., 2020). The slope of the regression line is equal to $-2 \cdot AEC(z)$ and is given by

$$AEC(z) = -\frac{1}{2}\frac{\partial Ln(ApBC(x,z))}{\partial x} \tag{1}$$

In this expression, the ApBC depends on both the horizontal distance to the aircraft $x$ and the flight altitude $z$. Since only measurements where the aircraft is not circling are retained, the angular stability with respect to the horizontal is better than 1°, which is equivalent to a vertically sampled layer of ~20 m. Only AECs associated with a relative regression error of less than 10% are retained. This avoids cloud-contaminated profiles in the regression range. The resulting error on the AEC is then lower than 0.01 km$^{-1}$. Given the ATR-42 flight strategy, AEC profiles in the lower troposphere were obtained across the trade wind inversion layer and the sub-cloud layer. For a given flight, the lidar-derived aerosol optical thickness (AOT) is computed by integrating the AEC along the altitude range covered by the aircraft $[z_b\ z_t]$ during the flight:

$$AOT = \int_{z_b}^{z_t} AEC(z) \cdot dz \tag{2}$$

with $z_t$ corresponding to an altitude a few hundred meters above the trade wind inversion and $z_b$ being located in the sub-cloud layer, a few tens of meters above the sea surface.

The mean VDR at a given altitude is also calculated over the same distance range along the lidar line-of-sight as the AEC:

$$\overline{VDR}(z) = \frac{1}{0.8}\int_{0.2}^{1} VDR(x,z) \cdot dx \tag{3}$$

The determination of the VDR from the lidar characteristics is described in Chazette et al. (2012). The absolute error on the VDR is close to 0.2%.

### 2.2.1 Assessment needing a calibration of the lidar signal

As opposed to AEC and VDR, LR and PDR calculations from horizontal shots require knowledge of the lidar system constant. It is assessed using the same linear regression as for determining the AEC but for the horizontal profiles at higher altitude where molecular scattering is predominant. The value at the origin of the regression function ($V_o$) is the product of the system constant (C) and the molecular backscatter coefficient (MBC), so that:

$$C = \frac{\overline{V_0(z)}}{MBC(z)} \tag{4}$$

Its value may change from flight to flight. However, as the lidar has remained in the same configuration, there is no reason for the system constant to change. Knowing $C$, in the presence of aerosols, the approach is similar and leads to the aerosol backscatter coefficient (ABC) by the relationship:

$$ABC = \frac{V_0(z)}{C} - MBC(z) \qquad (5)$$

The LR and PDR can thus be derived from the relationships (Chazette et al., 2012):

$$LR = \frac{AEC}{ABC} \qquad (6)$$

$$PDR(z) = \frac{MBC(z) \cdot (VDR_m - \overline{VDR}(z)) - ABC(z) \cdot \overline{VDR}(z) \cdot (1 + VDR_m)}{MBC(z) \cdot (\overline{VDR}(z) - VDR_m) - ABC(z) \cdot (1 + VDR_m)} \qquad (7)$$

where $VDR_m$ is the molecular volume depolarization ratio equal to 0.3945% at 355 nm (Collis and Russel, 1976).

## 3    Aerosol optical properties during EUREC⁴A

In this section, the optical properties of aerosols determined from horizontal lidar measurements are presented. The lidar calibration dependent variables and lidar calibration independent variables will be presented and discussed separately. Among the 20 flights conducted during the EUREC⁴A campaign (Chazette et al., 2020) we focus on the flights that are most representative of the different aerosol load observation periods:

- 28 January 2020 which corresponds to a background aerosol case, i.e. an atmospheric composition dominated by sea spray,

- 31 January 2020 which corresponds to the beginning of a 5-day period when aerosols other than sea spray were observed,

- 2 February 2020 when the maximum value of lidar-derived AOT was observed during the 5-day aerosol outbreak,

- 11 February 2020 which corresponds to a second aerosol outbreak event, less intense than the first one, during which aerosols other than sea spray were observed.

### 3.1    Directly retrieved aerosol optical parameters

On days when 2 ATR-42 flights were performed, the one with the most complete aerosol dataset is selected. Data from each of the 4 selected days are shown in Fig. 1 to 4, in chronological order (Figure 1 showing data from the flight performed on 28 January, etc…). The variability

of AEC and VDR along the flight path is shown in panels a and c of Figs 1-4. The corresponding AEC and VDR profiles (obtained after averaging all available data at a given altitude during the flight) are shown in panels b and d, respectively, of Figs 1-4. It should be noted that the vertical profiles include the ascent and descent parts of the flights. The rectangular pattern ABCD highlighted in Fig. 1a, was performed during each flights at an altitude of 700-800 m a.ms.l. corresponding approximately to the altitude of the cloud base (Bony et al., 2017; Chazette et al., 2020). The L-shape legs also seen in Fig. 1a were performed below cloud-base height, i.e. in the sub-cloud layer.

The background situation on 28 January (Fig. 1) is characterized by AECs generally below 0.1 km$^{-1}$ over the entire flight, with a lidar-derived AOT of ~0.08 at 355 nm. Higher values of AEC can be observed in the marine boundary layer (MBL) which is located here below 500 m a.m.s.l. and just above to the cloud base (~700-800 m a.m.s.l.). These values generally correspond to the highest values of relative humidity (~100%) and can therefore be associated with aerosol size growing due to its hydrophilic properties. The local source here is mainly oceanic. The vertical profile of AEC is characteristic of what is returned over the open ocean with aerosols mainly trapped in the MBL (Flamant et al., 2000). The VDR values show a presence of very weakly depolarizing aerosol which is therefore mostly spherical in shape.

The other three days show the presence of significant aerosol loads (Figs 2-4), mainly in the first two kilometres of atmosphere, i.e. below the trade winds inversion height. The AEC exceeds 0.2 km$^{-1}$ in some parts of the profiles and the VDR is significantly increased compared to 28 January. This could be the signature of dust-like aerosols, which are observed up to ~2.5 km a.m.s.l. on 2 February. The vertical profiles show different structures on the selected 3 days, with a higher AEC in the MBL on 31 January while the particles seem to be more vertically homogeneously distributed on 2 February (Fig. 3c-d) with an AEC maximum just below the trade winds inversion. The vertical profile on 11 February highlights 2 distinct layers, with AEC maxima in both the MBL and above, below the trade winds inversion. The significance of the vertical variabilities for the AEC and VDR is assessed by comparing the point-to-point statistical variability (light grey area in Figs 1-4) and the statistical error on the mean value (dark grey area). The point-to-point variability includes the variability due to the atmospheric environment and the variability due to the measurement noise. We thus have a very good assessment of the mean vertical profiles.

What emerges from such atmospheric sampling is that it reveals significant small-scale horizontal heterogeneity in the optical properties of aerosols. This heterogeneity can be related

to the variability of the horizontal relative humidity field if the particles are partly hydrophilic, but also to contributions from different sources, or even to convective processes on the scale of the clouds which are present at this period.

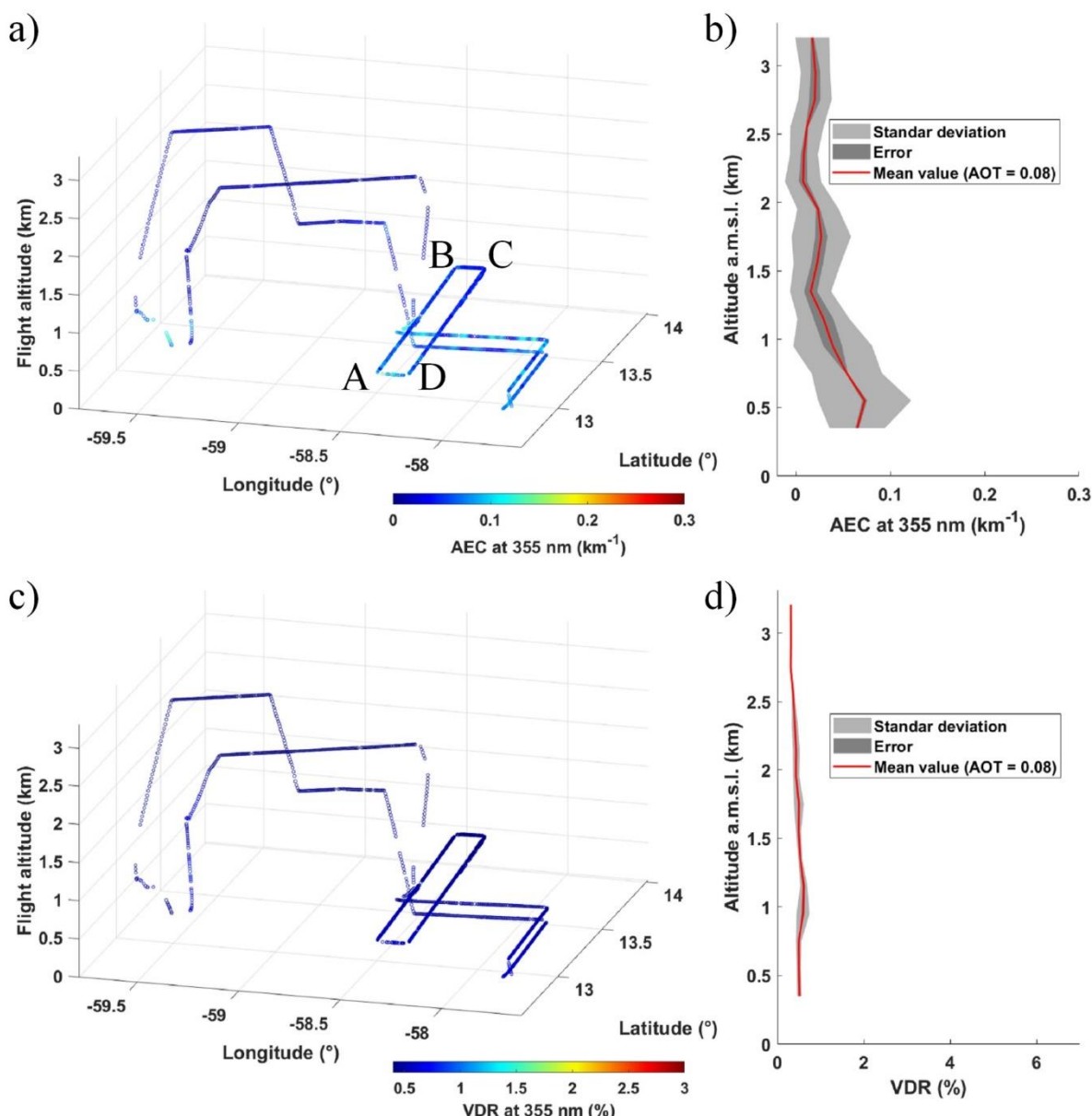

Figure 1. a) Aerosol extinction coefficient (AEC) derived from ALiAS measurements during the flight F05 on 28 January 2020 1615-2050 UTC along the horizontal line of slight, b) corresponding vertical profile of the AEC which shows the spread of measurements via the standard deviation (light grey area) and the statistical error (dark grey area). The same types of figures are given for the linear volume depolarization ratio (VDR) in c) and d), respectively. The level where a rectangle ABCD has been systematically depicted is highlighted in a).

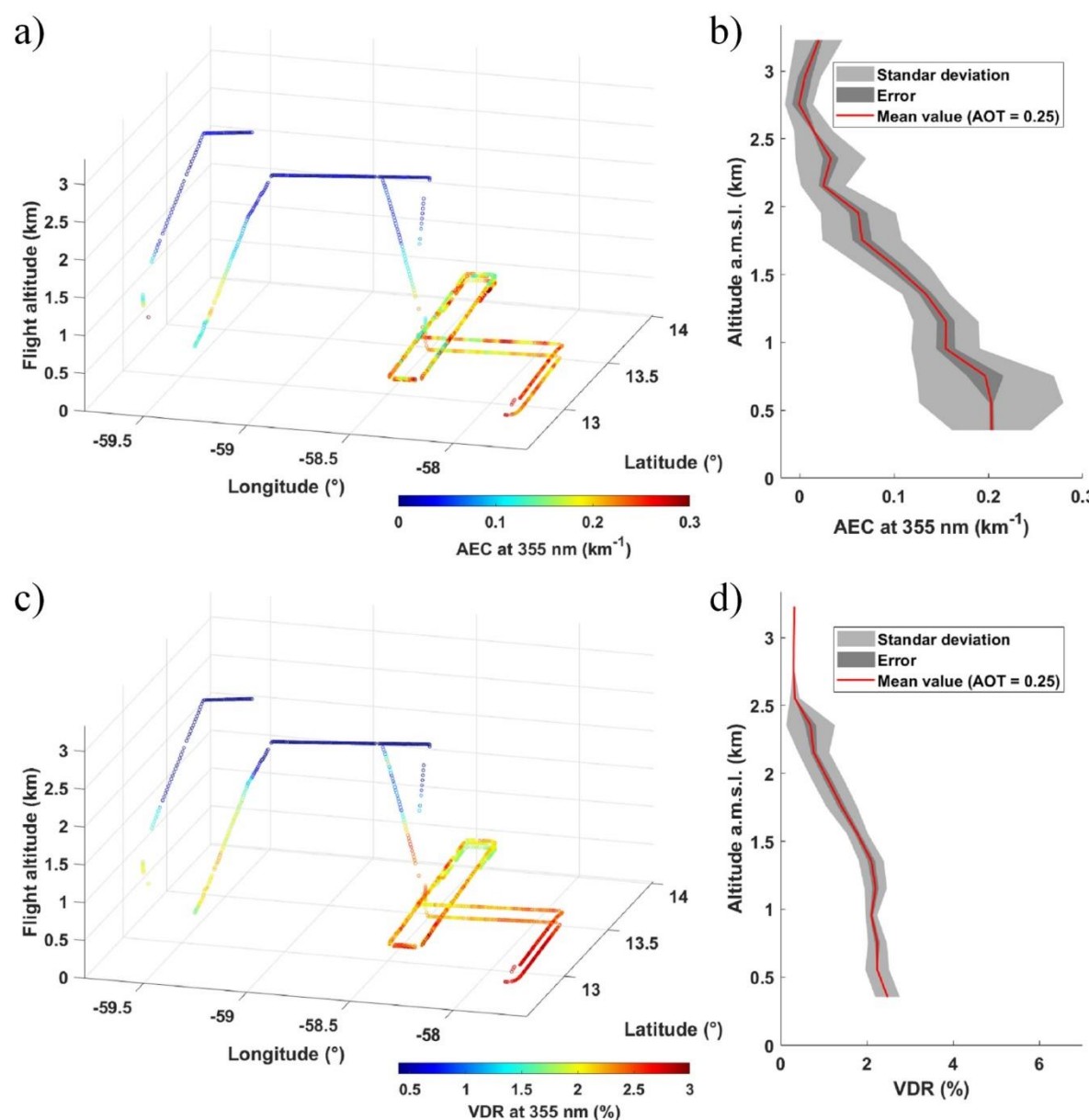

Figure 2. Same as Fig. 1 during flight F08 on 31 January 2020 1945-2400 UTC.

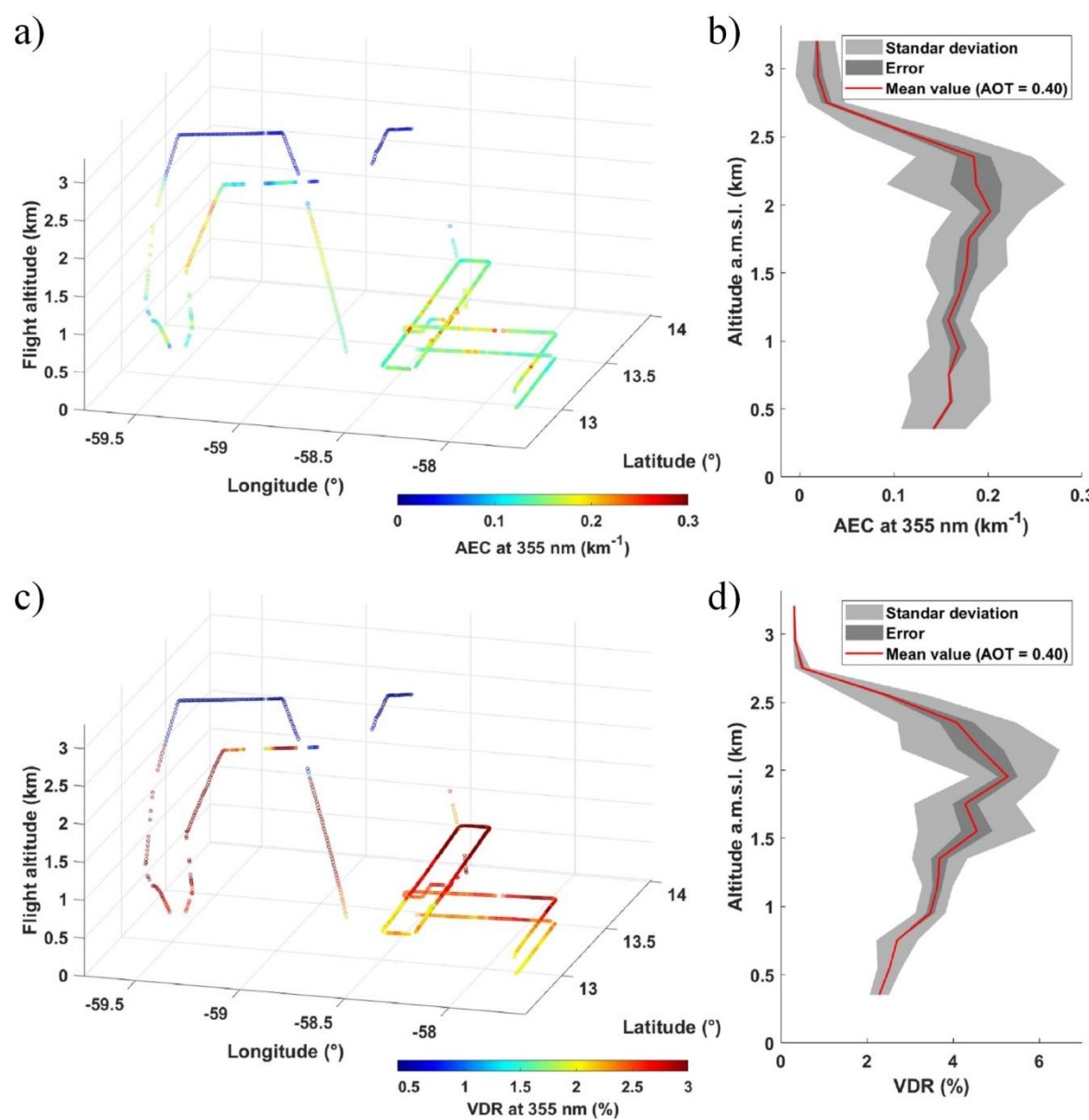

Figure 3. Same as Fig. 1 during flight F10 on 2 February 2020 1645-2100 UTC.

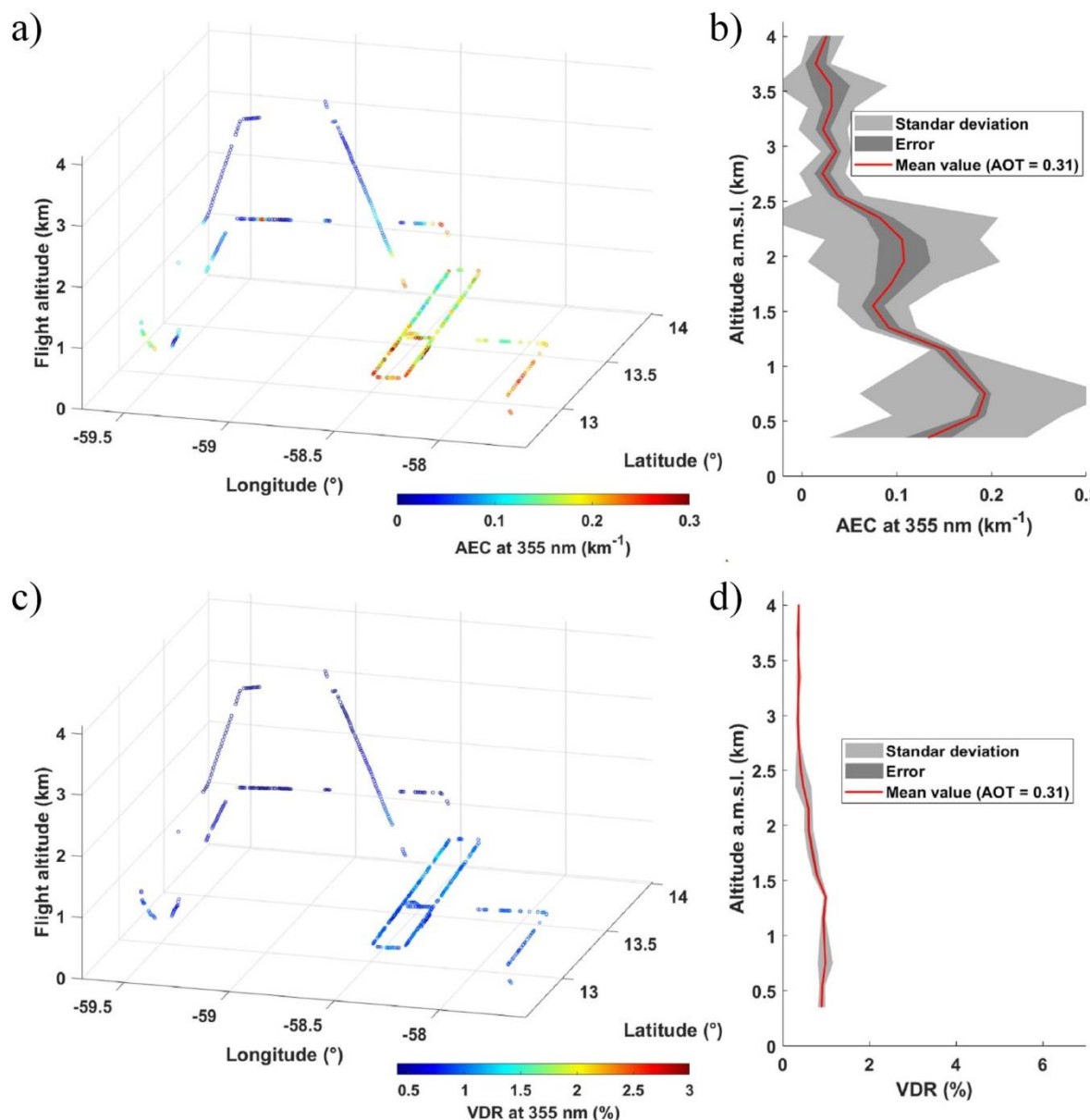

Figure 4. Same as Fig. 1 during flight F18 on 11 February 2020 1130-1600 UTC.

### 3.2    Optical parameters requiring calibration

As discussed above, LR and PDR retrievals require calibration of the lidar data. This calibration
5    is flight-dependent and is not due to the intrinsic lidar settings, which are stable during the
flights (pressurised and temperature-controlled cabin), but to the evolution of the smearing on
the side window. This smearing is related to the presence of aerosols and is very often strongly
accentuated when the aircraft has flown in the MBL where sea spray causes salt deposits, which
then agglomerate with other types of aerosols during the flight. Among the 20 flights conducted
10    during EUREC[4]A, LR and PDR calculations could only be made on 31 January and 2 February
2020.

In the case of 31 January, heavy smearing was encountered during the evening flight (the second flight of the day, see Table 5 of Chazette et al. (2020)), i.e. for flight F08 shown in Fig. 2, so that LR and PDR on that day were determined from data acquired during the morning flight (flight F07, see Table 5 of Chazette et al. (2020)). The average vertical profiles of flight F07 are presented in Fig. 5. The vertical profile of AEC is significantly different than the one in Fig. 2b although the AOT remains of the same order. There is a stronger spread in the data due to the high cloudiness at the flight altitude. The VDR remains around 2% at the bottom of the profile. The LR calculation leads us to values of the order of 45±10 sr corresponding to dust aerosols or a mixture including dust aerosols (Burton et al., 2012). The PDR is not very high and remains around 5% in the aerosol layers, both in the MBL and above. This effectively argues for a mixture of different aerosol types (Chazette et al., 2016) including dust particles.

LR and PDR derived on 2 February (flight F10) are shown in Fig. 6. The heterogeneous nature of the aerosol layers for a given flight stage can be noted. The LR ranges from 40 to 60 sr is always characteristic of a dust mixture. The PDR is higher than on 31 January, which shows a stronger influence of dust particles in the aerosol mixture. Moreover, this mixture is found up to an altitude of 2.5 km a.m.s.l., as the vertical profile in Fig. 6d clearly shows. As previously stated, the variability observed in LR and PDR could be related to the hydrophilic character of the aerosol mixture. Nevertheless, LR does not rapidly evolve with increasing RH as shown by Raut and Chazette (2007). Therefore, aerosols of different nature could have been sampled during the flight.

During the first dust transport event (i.e. on 31 January and 2 February), the vertical extent of the dust-BB layer (~2 km a.m.s.l. on average) as well as the PDR values on the order of 5-10% are found in good agreement with Haarig et al. (2017) over Barbados, except that in our case, the mixed aerosol layer is well-mixed from the surface. On the other hand, the LR in our case was derived to be much larger, i.e. 40-60 sr, than in Haarig et al. (2017) who found values ranging from 20 to 30 sr at 355 nm in February 2014.

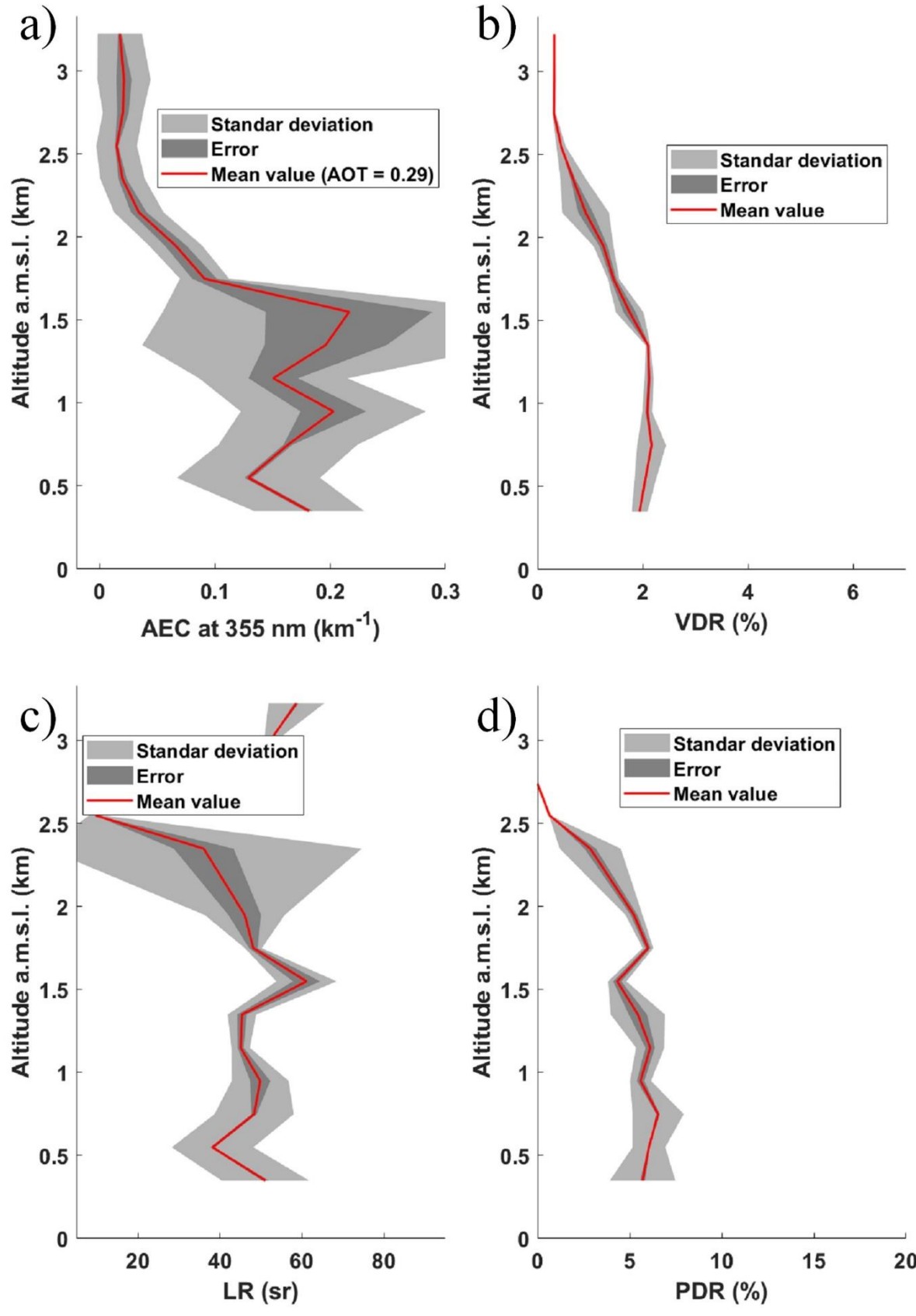

Figure 5. Vertical profiles derived from ALiAS during flight 07 on 31 January 2020 1500-1845 UTC of a) the aerosol extinction coefficient (AEC), b) the volume depolarization ratio (VDR), c) the lidar ratio (LR), and d) the particle depolarization ratio (PDR).

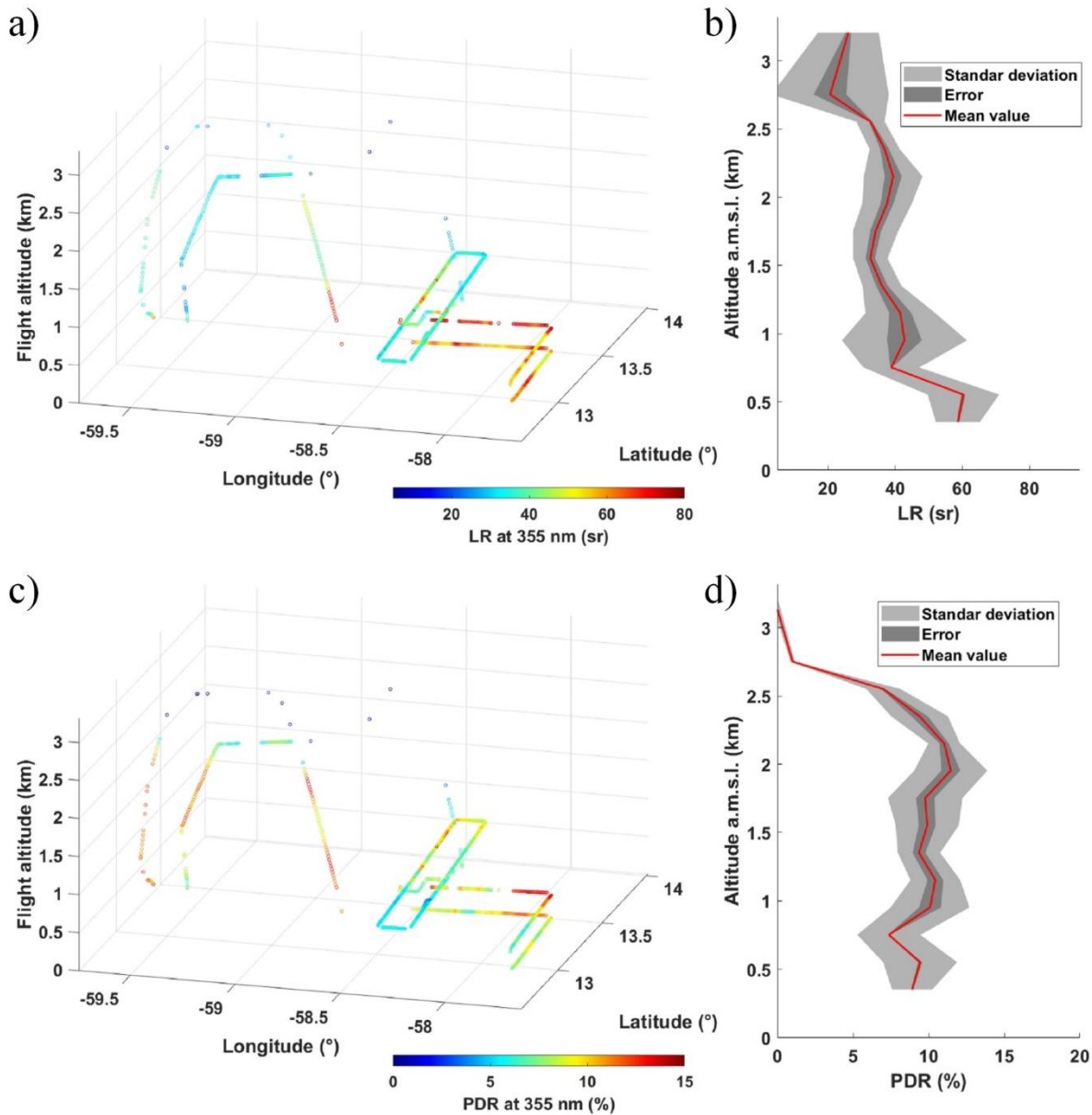

Figure 6. a) Lidar ratio (LR) derived from ALiAS measurements during the flight F10 on 2
February 2020 1945-2100 UTC, b) corresponding vertical profile of the LR which shows the
spread of measurements via the standard deviation (light grey area) and the statistical error
(dark grey area). The same types of figures are given for the particle depolarization ratio (PDR)
in c) and d), respectively.

## 4   Large-scale spatiotemporal evolution

The identification of the origin of the aerosols, which allows to confirm or deny the previous

results, can be done by using larger scale observations in order to visualize the whole tropical

Atlantic basin, from Africa to the Caribbean Sea. The Moderate-Resolution Imaging

Spectroradiometer (MODIS) (King et al., 1992; Salmonson et al., 1989) and the Cloud-Aerosol

Lidar with Orthogonal Polarization (CALIOP) (Kim et al., 2018; Winker et al., 2007) space-

borne observations are used, together with atmospheric composition outputs from the

Copernicus Atmosphere Monitoring Service (CAMS, https://atmosphere.copernicus.eu/, last

access: 17 December 2021). To complement space-borne observations and aerosol transport modelling, 11-day back trajectory analyses are performed with the Hybrid Single Particle Lagrangian Integrated Trajectory (HYSPLIT) model (Draxler and Rolph, 2014; Stein et al., 2015). The wind fields used are from the Global Forecast System (GFS) of the National Centers for Environmental Prediction (NCEP) weather forecast model at 0.25° horizontal resolution. The isentropic ensemble mode with 24 individual back trajectories is used to take into account the transport trajectory spread.

### 4.1  3D distribution derived from MODIS and CALIOP measurements

For the 3 days where the ALiAS lidar measurement showed aerosol loads significantly different from the background content (31 January, 2 and 11 February), the MODIS and CALIOP observations were extracted and are shown in Figs. 7-9. For all 3 days, aerosol plumes are observed and extend from the African coast to the Caribbean Sea (see Figs 7a, 8a and 9a). The CALIOP-derived three-dimensional distribution of AEC (Figs 7b, 8b and 9b) as well as aerosol classification (Figs 7c, 8c and 9c) show that these plumes are generally observed between sea level and 2.5-3 km a.m.s.l., i.e. below the trade winds inversion, and this throughout the Atlantic. Obviously, this is a snapshot taken over one day and these figures do not fully reflect the dynamics of the phenomenon. Analysing the origin of the aerosol plumes on the MODIS data and the identification of the aerosol type (dust and polluted dust) from the CALIOP measurements, we see that the aerosols observed near Barbados are most likely mixtures of dusts and biomass burning aerosols, as already evidenced by Haarig et al. (2017, 2019) during the winter campaign of SALTRACE, which certainly also include sea salts and sulphates. The main sources appear to be mainly located on the African continent and to a lesser extent in South America.

The MODIS fire product shows that the sources of biomass burning aerosol are widespread during this season between Guinea Bissau and Ivory Coast (Giglio et al., 2006), territories that seem to be the origin of a significant part of the plume. The sources of dusts seem to be located at the latitude of Mali and Occidental Sahara. CALIOP identifies dusts along the west coast of Africa, from Senegal towards Morocco. The aerosol mixing that can be inferred along the transport across the Atlantic confirm the results obtained from the ATR-42 lidar measurements.

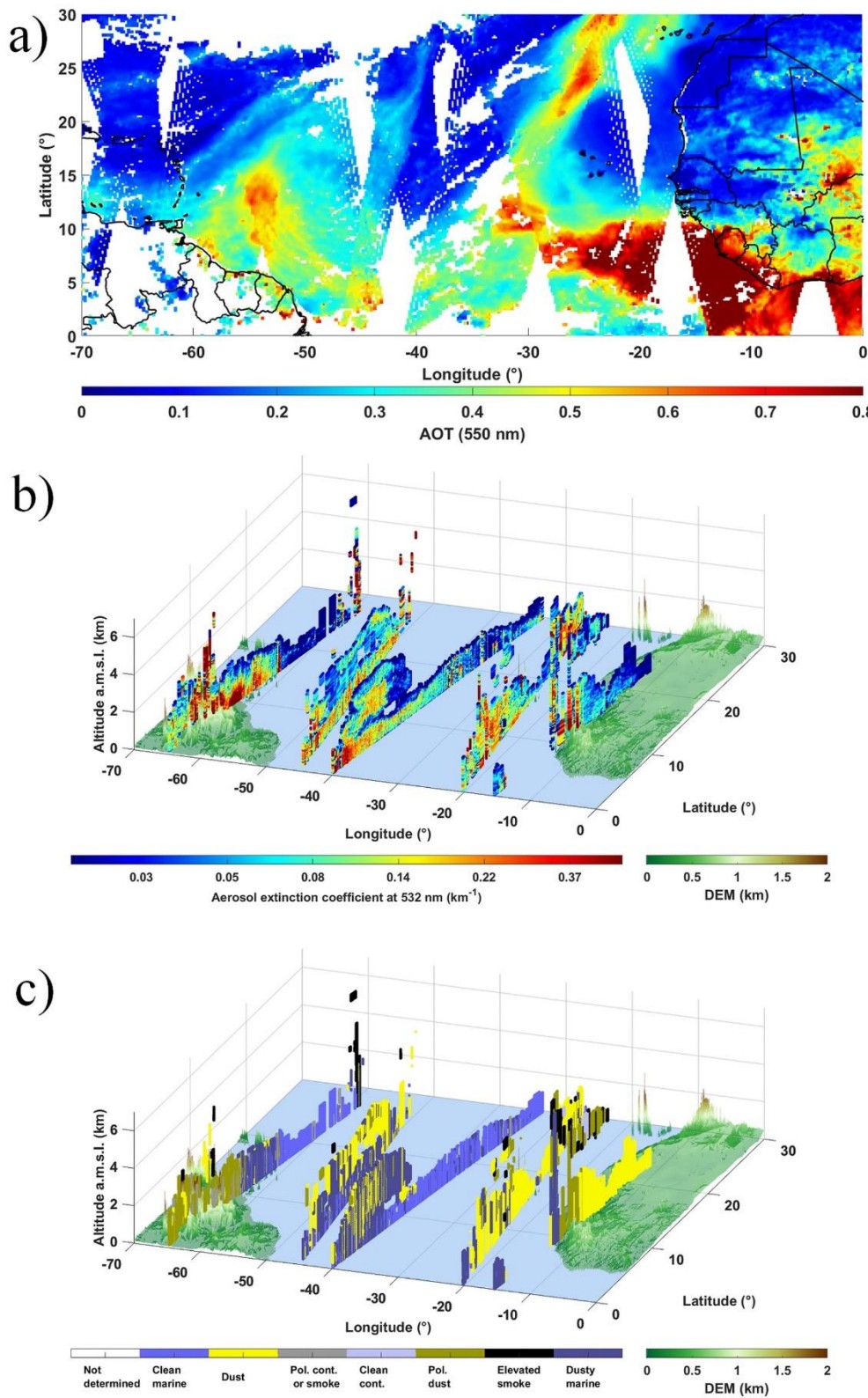

Figure 7. a) MODIS-derived AOT at 550 nm, b) CALIOP aerosol extinction coefficient at 532 nm for all orbits, and c) CALIOP aerosol classification for all orbits on 31 January 2020, in V4.1 version of the operational algorithm.

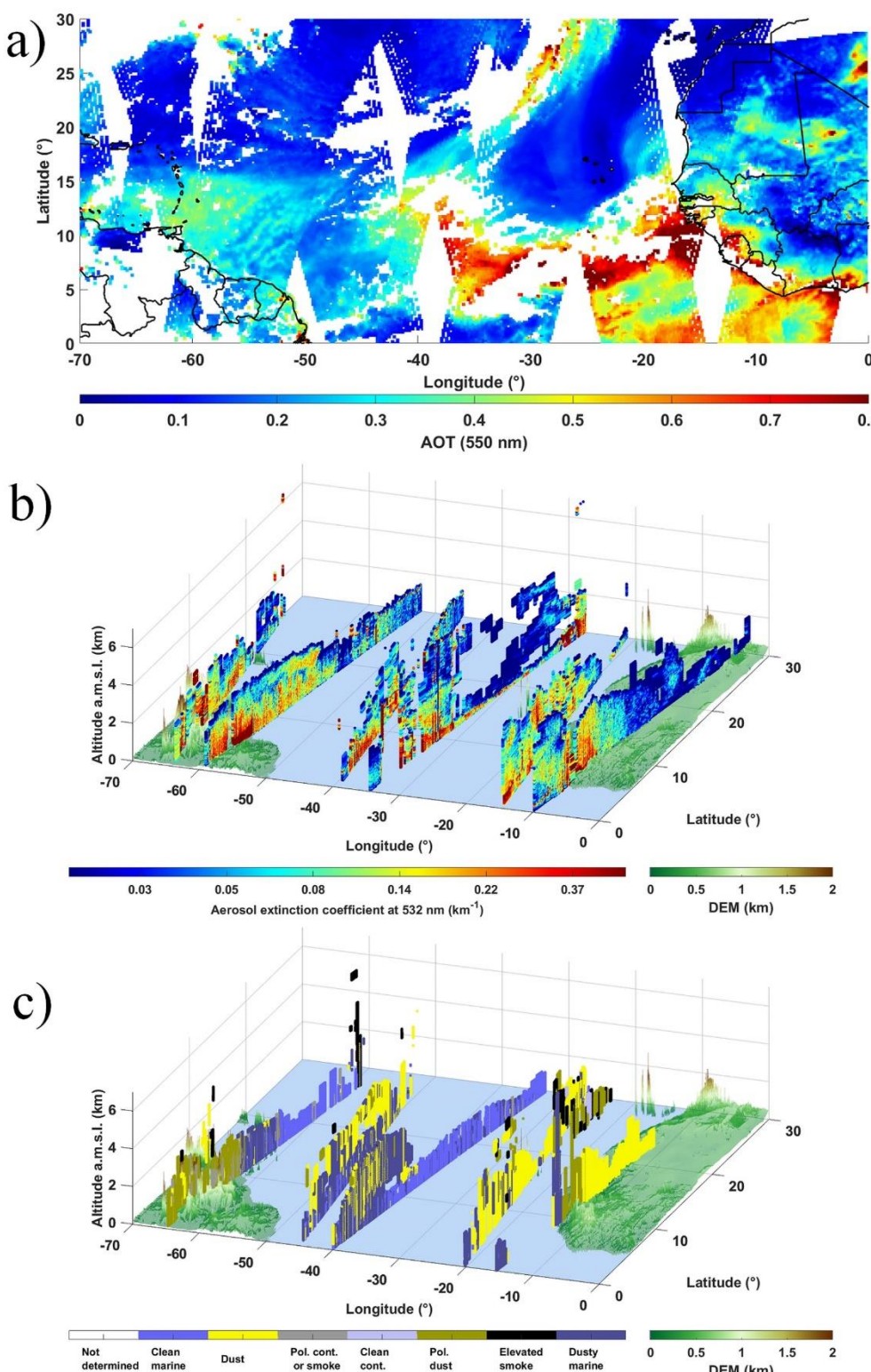

Figure 8. Same as Figure 7 but on 2 February 2020.

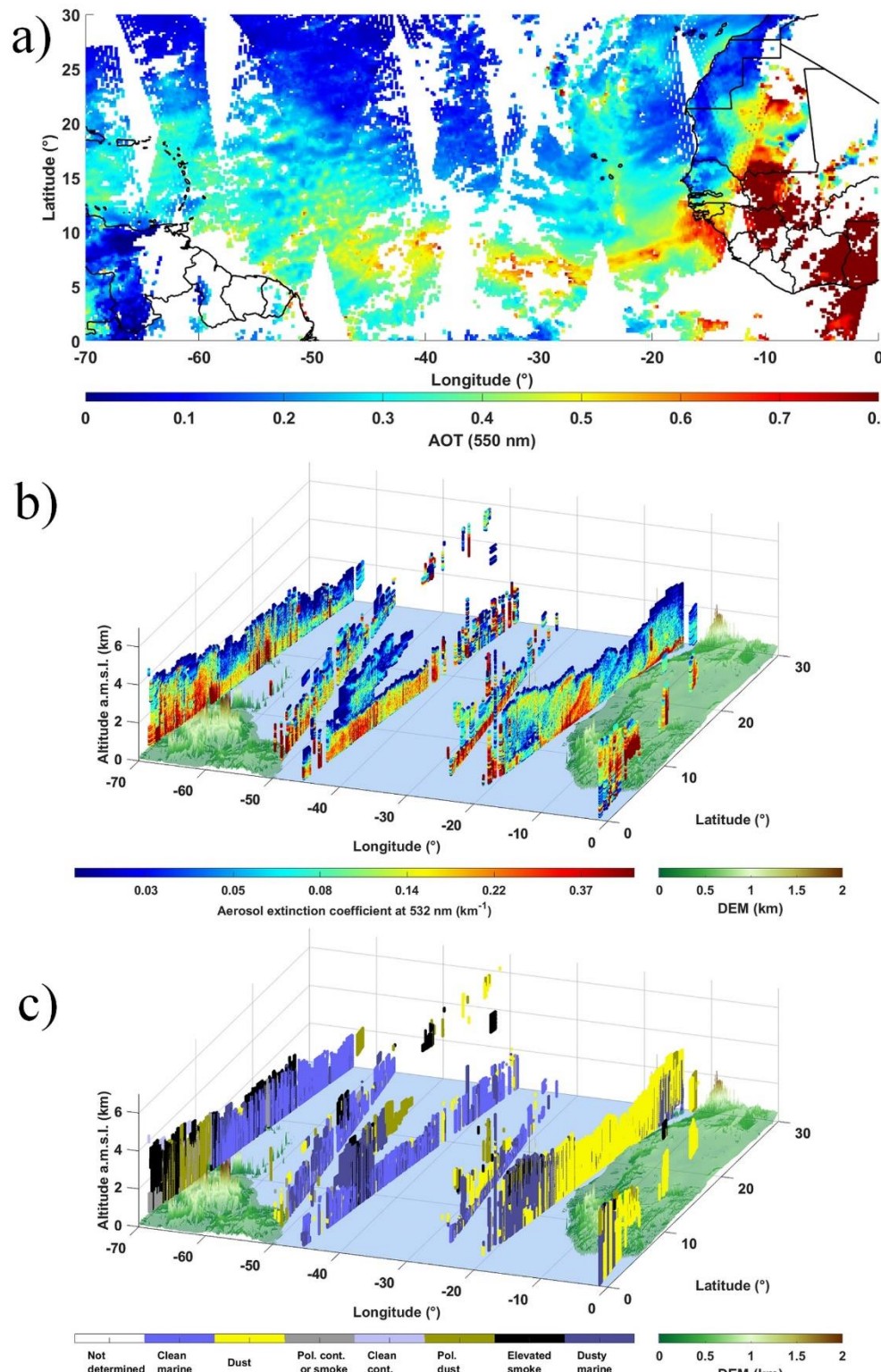

Figure 9. Same as Figure 7 but on 11 February 2020.

## 4.2 Horizontal distribution of aerosol plumes via CAMS

CAMS numerical simulations presented in Figs. 10-12 confirm the previous conclusions. As
5    CAMS assimilate the MODIS-derived AOT (Benedetti et al., 2009; Inness et al., 2018), the
aerosol plume reproduced by the model is very close to the one actually observed by MODIS.

However, the advantage of the CAMS model is that it provides the chemical and optical speciation of the aerosols in the plume. For the three previous days, the two main aerosol components identified are dusts and carbonates (organic carbon and soot carbon), the latter being naturally emitted by biomass burning. The contribution to the AOT of the two types of compounds is almost equivalent. The simultaneous presence of dusts and biomass burning aerosols may explain the heterogeneous character observed above. This does not exclude the role of relative humidity as explained by Kim et al. (2009) for the winter period in West Africa. They have shown that the biomass burning aerosol plumes advected over long distances are associated with significantly higher relative humidity values than the dust plumes. These two plumes may co-exist at different altitudes or be mixed as in our case. This mixture may not be homogeneous.

As with the satellite observations, the daily maps do not provide information on the dynamics of aerosol movement reaching Barbados. For this reason, we computed back trajectories by initializing them at the altitudes of the aerosol layers located by the airborne lidar measurements, the lowest initialization altitude corresponding for each flight to the level where the rectangle ABCD was described (Fig. 1a), i.e. 700-800 m a.m.s.l.

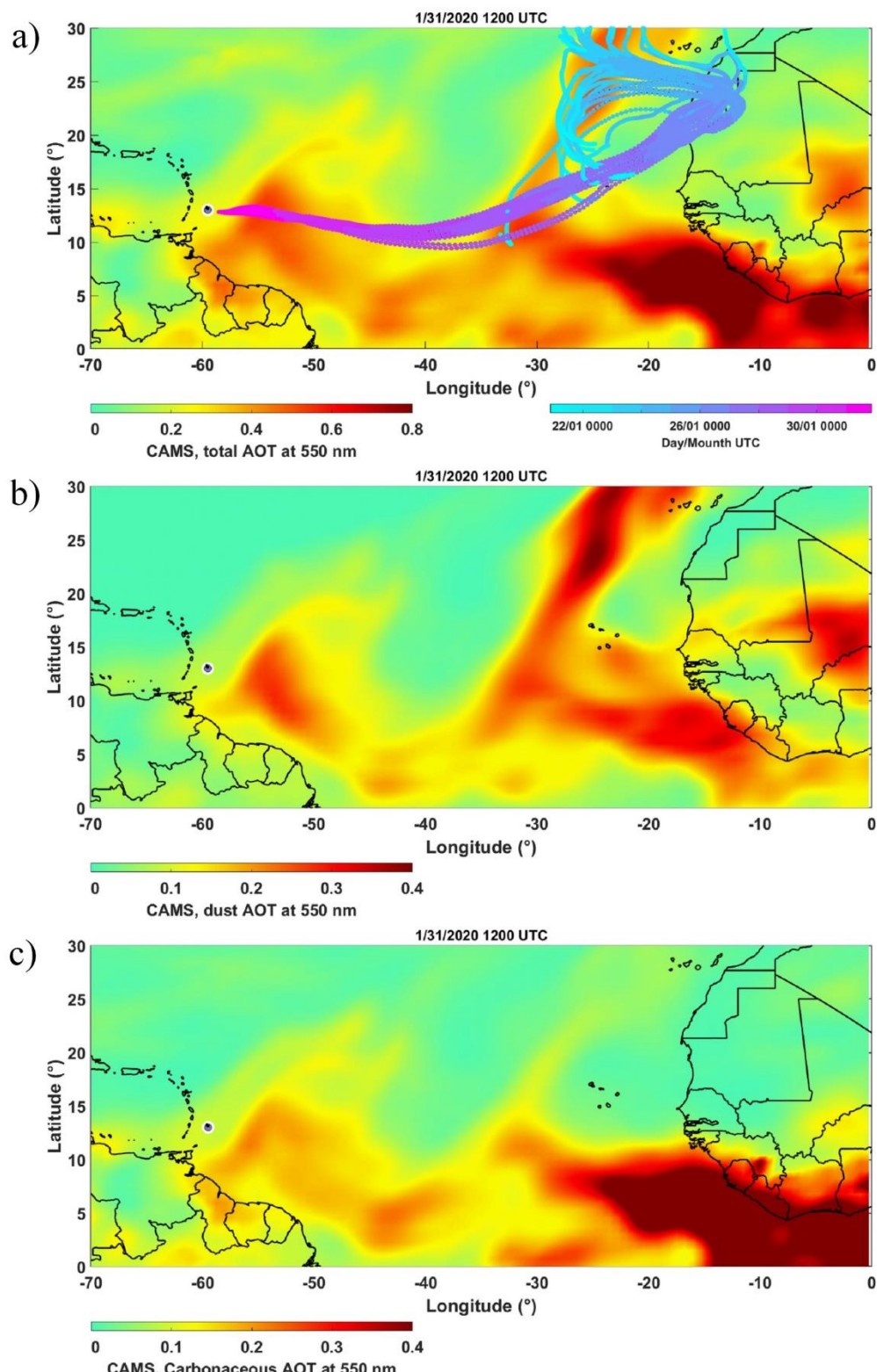

Figure 10. Aerosol optical thickness (AOT) derived from the CAMS numerical simulations on 31 January 1200 UTC: a) total AOT, b) AOT of dusts, and c) AOT of carbonaceous components (black carbon and organic carbon). The back trajectories computed via Hysplit in ensemble mode are plotted in a) for an initial altitude of 700 m a.m.s.l. over the flight area close to Barbados, corresponding to rectangle ABCD in Fig. 1a.

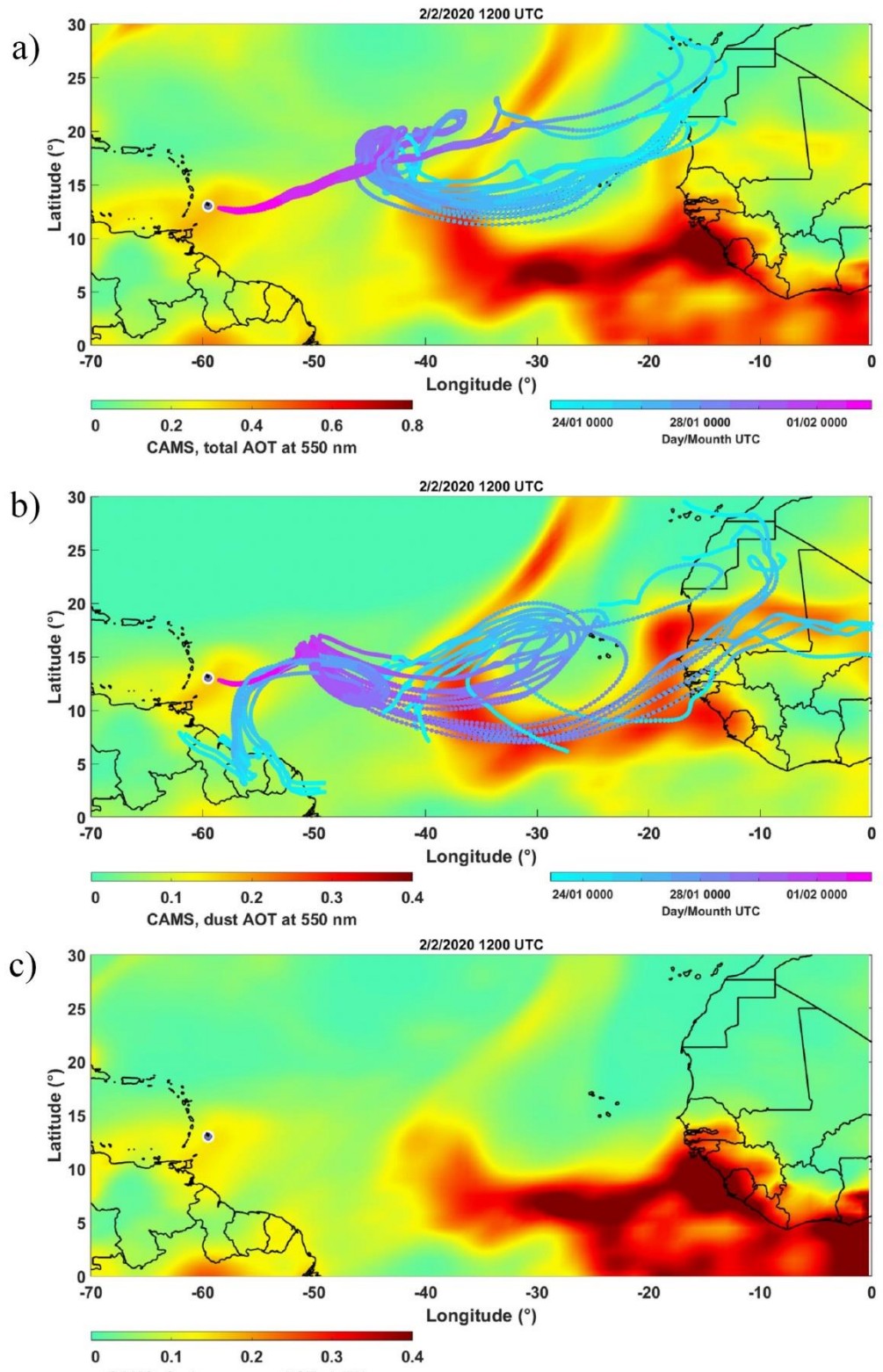

Figure 11. Aerosol optical thickness (AOT) derived from the CAMS numerical simulations on 2 February 1200 UTC: a) total AOT, b) AOT of dusts, and c) AOT of carbonaceous components (black carbon and organic carbon). The back trajectories computed via Hysplit in ensemble mode are plotted in a) for an initial altitude of 800 m a.m.s.l., corresponding to rectangle ABCD in Fig. 1a, and in b) for an initial altitude of 2250 m a.m.s.l., both over the flight area close to Barbados.

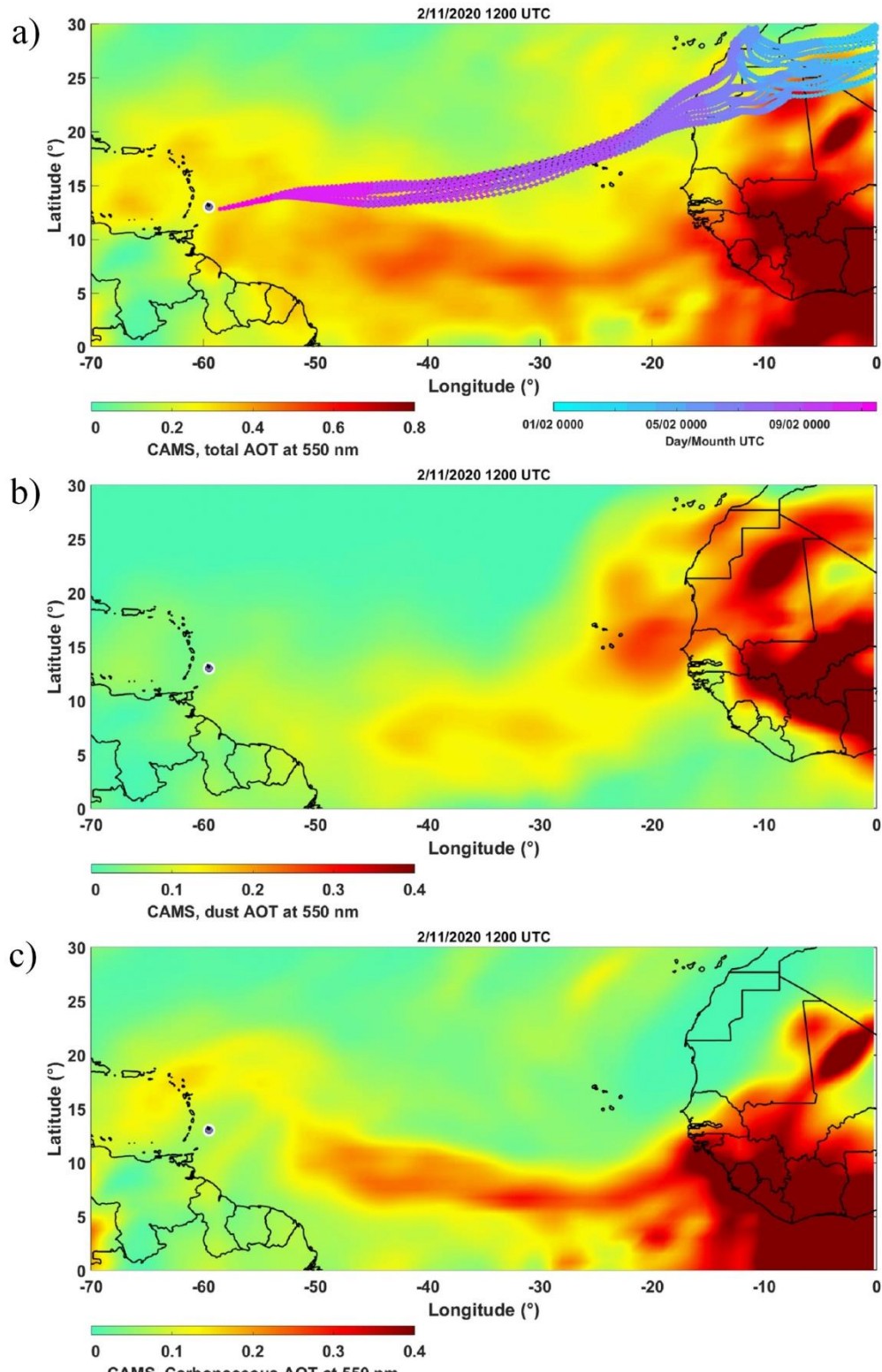

Figure 12. Aerosol optical thickness (AOT) derived from the CAMS numerical simulations on 11 February 1200 UTC: a) total AOT, b) AOT of dusts, and c) AOT of carbonaceous components (black carbon and organic carbon). The back trajectories computed via Hysplit in ensemble mode are plotted in a) for an initial altitude of 750 m a.m.s.l. over the flight area close to Barbados, corresponding to rectangle ABCD in Fig. 1a.

## 4.3 Back trajectory analyses

On 31 January, the initial altitude of the back-trajectories was chosen at 700 m a.m.s.l. where the airborne lidar provided the best sampling just above the MBL. Fig. 10a shows the back-trajectories in ensemble mode. They are all clustered over the Atlantic Ocean and pick up the dust plume off the Western Sahara and Mali. They then cross the biomass burning plume at about 30°W. These air mass trajectories are all located below 1.5 km a.m.s.l. when crossing the Atlantic Ocean. For February 2 (Fig. 11), the scenario is a bit more complex. Already, two levels of departure altitude of the back trajectories have been considered, 800 and 2250 m a.m.s.l., to take into account the vertical extent of the plume as shown in Fig. 3. In the lower layer, the particles observed seem to originate from the coast of the Western Sahara, whereas in the higher layer they have various origins. Indeed, they may have originated from Malian sources for the dust aerosols and mixed African and South American sources for the biomass fire aerosols. The mixture seems to be quite complex as the aerosols are all entrained in the central Atlantic before being transported over Barbados. On 11 February, the back-trajectories initiated at 750 m a.m.s.l. are very close and seem to indicate mainly a dust source located in northwest Africa whose plume could be less mixed with biomass burning aerosols.

### 4.4    Vertical aerosol speciation as derived from CAMS

Figure 13a shows the contribution to the AOT of the different aerosol types over time for an atmospheric column located in the centre of rectangle ABCD (Fig. 1a). The flights considered in this paper are marked by red dotted lines while the other flights are highlighted by dark grey dotted lines. It is worth noting that the selected observation periods are representative of the main aerosol situations, with rather contrasting relative contributions of aerosol compositions. The major contributions are related to dust and carbonaceous aerosols. Sea salt and sulphate aerosols contribute less than 10%. An exception is for 11 February where all aerosol types have about the same contribution. This can be explained by higher ocean surface wind speeds (> 10 m s$^{-1}$) which favoured the suspension of marine aerosols (Blanchard and Woodcock, 1980; Flamant et al., 2000). Such mixing does indeed lead to a decrease in depolarization as observed via the VDR, as more spherical hydrophilic particles may be present.

Figs. 13b-c show the corresponding vertical distributions of dust and carbonaceous aerosols, respectively. There is a very good agreement between the flight-derived lidar vertical profiles, and the vertical structures reproduced by CAMS. It is noteworthy that the vertical distribution of aerosols is not constrained in the assimilation process, as opposed to the horizontal distribution of aerosols. The mass contribution of dust aerosols is 5 times higher than that of carbonaceous aerosols, but the specific extinction cross section of the latter compensates for

this difference (Raut and Chazette, 2009). On 2 February, dust and carbonaceous aerosols are distributed in the same proportions according to altitude, which explains the almost constant PDR observed up to ~2.5 km a.m.s.l. (Fig. 6d). The maximum altitude concentration of both compounds (~2.2 km a.m.s.l.) corresponds effectively to the AEC maximum in Fig. 3a.

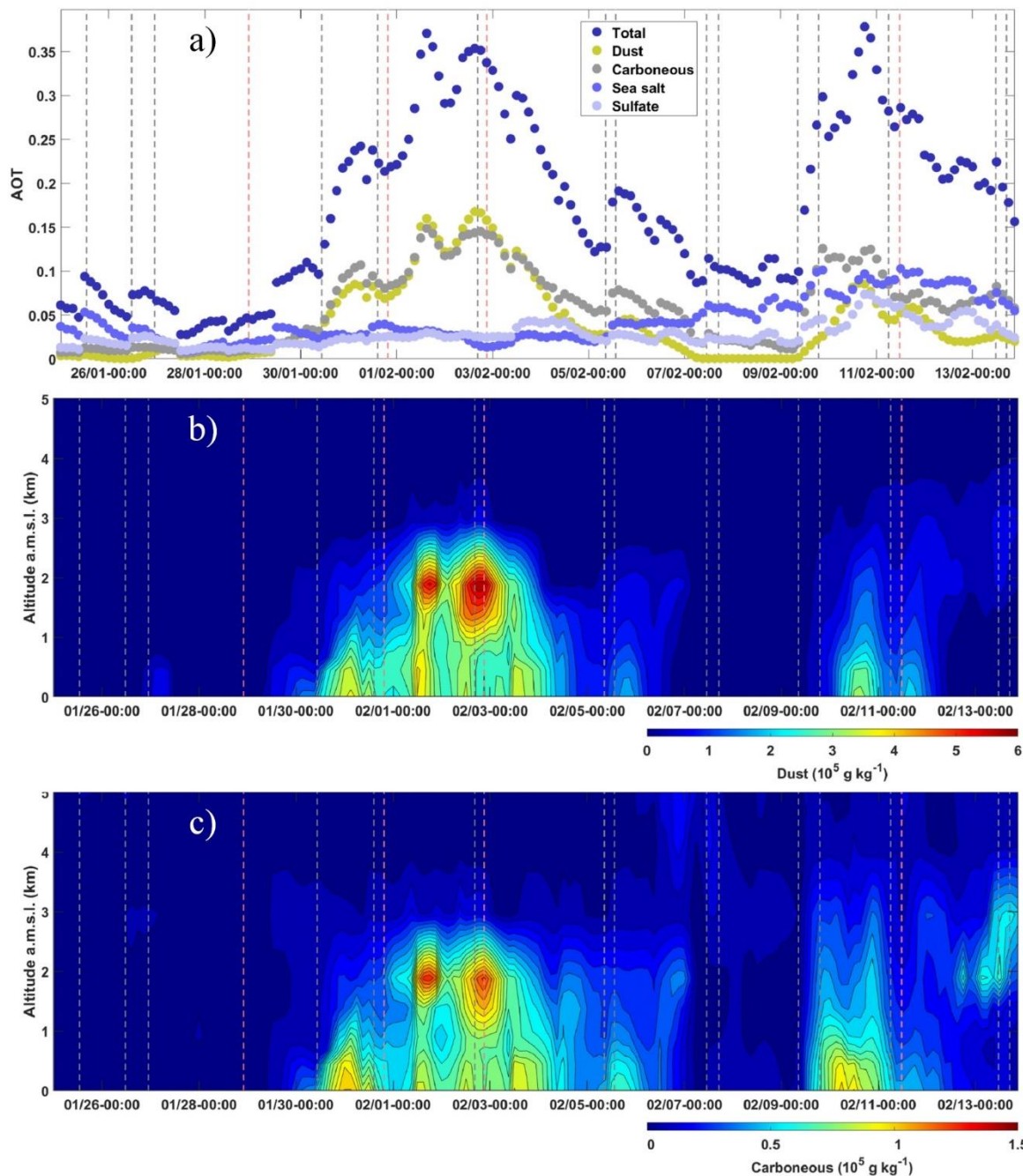

Figure 13. Temporal evolution derived from CAMS of a) the aerosol optical thickness (AOT) at 550 nm for different aerosol compounds, b) the vertical profile of dust aerosol mass concentration and c) the vertical profile of carbonaceous (black carbon and organic carbon) aerosol mass concentration. The locations of the considered flight are highlighted by red vertical dotted lines. The other flights are highlighted by black-grey vertical dotted lines.

## 5   Discussion - Relationship to transport and weather conditions

The intensity of long-range transport and even the contribution of aerosol sources observed during EUREC[4]A are closely linked to transitions between different weather regimes. Until 27 January 2020, the Azores High is positioned over the eastern Atlantic (centred on 25°W) while a low pressure is located over the western Atlantic (centred on 55°W), north of the Caribbean,

and the westerly sub-tropical jet is positioned north of 45°N. As a result, dust outbreaks from northern West Africa are seen to travel westward along 20°N over the Atlantic before being redirected towards the northeast due to the strong south-westerly flow between the low-pressure centre and the Azores high and cannot reach Barbados. Between 27 and 30 January, both the low-pressure centre and the high-pressure system moves eastward, the former across the

Atlantic and the latter over North Africa (leading to the air mass trajectories see in Fig. 10a off the coast of Africa), as a result of an equatorward undulation of the westerly sub-tropical jet and an associated deep low trough off the US east coast. At this stage, the westerly flow associated with the high-pressure centre over northern Africa is still not strong enough to reach the Caribbean. From 31 January 1200 UTC onward, a high-pressure centre is travelling along

30°N north of the Caribbean while the high-pressure system over north Africa strengthens. The aerosols transported out of the African continent over the Atlantic by the circulation around the easternmost anticyclone are then picked up by the easterly circulation south of the westernmost high-pressure system and can now reach Barbados. As the western high pressure moves westward, aerosols out of West Africa are more efficiently transported across the Atlantic

towards Barbados from 1 to 4 February 2020. During this period, 2 cut-off lows are also seen to travel eastward in between the 2 high pressure systems, one of which is shown in Fig. 14a on 2 February. Rossby wave breaking events are also occurring ahead of the cut-off lows. The presence of the 2 distinct high-pressure centres (and the lower pressure in between) leads to the complex aerosol recirculation (loop) evidenced in the back trajectories ending in Barbados on

2 February (Fig. 11a, b). It may also be the case that the injection of both dust and carbonaceous species above the sub-cloud layer and below the trade winds inversion is related to the interactions between the 3 features over the Atlantic.

After this first episode of transport, the high-pressure system over North Africa is seen to move northward over Europe. Subsequently, between 7 and 13 February, an elongated high-pressure

system develops across the whole Atlantic, along 30°N, which favours direct transport of African aerosols towards the Americas. At first, the intensity of transatlantic ridge partially blocks the transport of aerosol towards the Barbados region (7-9 February), then as it moves northward (see Fig. 14b) air masses from West Africa are able to reach the Caribbean following

a very direct course (Fig. 12a), as opposed on the 2 February case. Furthermore, the well-established transatlantic high-pressure and the easterly circulation to its south create near-surface wind conditions favourable to the significant, steady production of sea spray and sulphates that are seen in CAMS atmospheric composition near Barbados (Fig. 13). These two very contrasting situations are to be related to the weather patterns identified by Aemisegger et al. (2021). The situation on 2 February corresponds to the trade wind regime which favours the export of dusts via the recirculation of particles raised above very active sources such as those of the Bodélé. The case of 11 February is associated with the tropical regime and seems to favour coastal sources such as those of the Western Sahara and Mali.

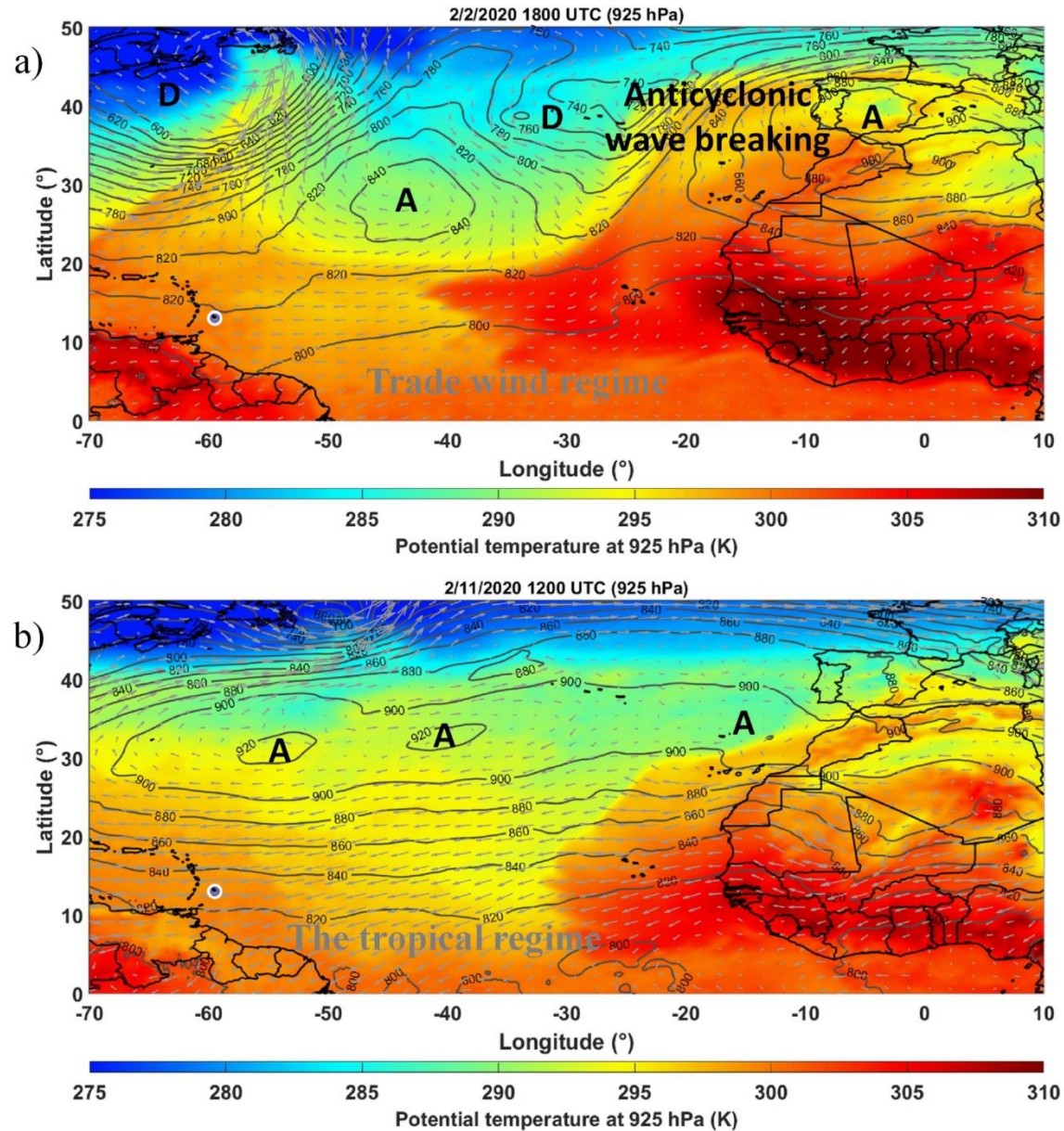

Figure 14. Equivalent potential temperature at 925 hPa on a) 2 February 1800 UTC, example of the trade wind regime with an anticyclonic Rossby wave breaking and b) 11 February 1200 UTC, example of the tropical regime. The anticyclones (A) and depressions (D) are indicated.

## 6    Conclusion

This study has shown a very strong consistency between airborne lidar observations, passive and active satellite instrumentation. All these measurements are also in very good agreement with the CAMS aerosol reanalysis products, and in particular with regard to the vertical distribution of aerosols. The aerosol loads sampled over Barbados during EUREC[4]A, and to a lesser extent the nature of the aerosols, are shown to be closely related to the weather patterns encountered. It is usually assumed that dust is transported towards northeast South America in January-February, i.e. south of the Caribbean, and that the tropical Atlantic towards North America is relatively dust free. We show that this is not always the case, depending on the winter transport regime from West Africa. The trade wind regime favours the export of dust and biomass burning from equatorial Africa. The complex interactions with the mid-latitude dynamics are responsible for the transport of fairly deep (still less so than in summertime), well mixed aerosol layers towards the Caribbean. This regime can even induce by recirculation the transport of biomass burning aerosols from South America. The tropical regime favours more direct transport of aerosols from sources along the west coast of Africa, as well as sea spray (due to the higher near-surface wind speeds) in shallower layers than associated with the trade wind regime. Three-dimensional aerosol fields appeared to be highly variable in time, but also in space at the scale of airborne measurement, i.e. over distances of a few tens of kilometres. This is difficult to capture by other types of observations which are either very local or tend to apply a low pass filter on the three-dimensional fields. This variability may be related to transport processes, but most likely to dynamical processes associated with the presence of fractional cloud fields that influence the three-dimensional distribution of aerosols through the convection it generates at the top of the MBL and through the strong variability of relative humidity on flight levels. Such a heterogeneity of the aerosol field could significantly modulate the climatic impact of aerosols trapped over the tropical Atlantic. The EUREC[4]A project had as its main objective the study of clouds and their link to equatorial and subequatorial dynamics, but it was also a unique opportunity to help characterise the optical properties of aerosols and study their transport across the tropical Atlantic. Our wintertime airborne lidar observations over Barbados corroborate the few existing studies of dust-BB aerosol mixture optical properties in the area by Haarig et al. (2017, 2019).

**Data availability.** The airborne lidar ALiAS datasets are published open access on the AERIS database (https://en.aeris-data.fr/, last access: 17 December 2021). The digital object identifier (DOI) is https://doi.org/10.25326/59 for the ALiAS-derived aerosol products during EUREC[4]A.

**Author contributions.** Patrick Chazette participated to the field experiment on board ATR-42 as PI of lidar measurements, he analysed the data, and wrote the paper; Alexandre Baron and Cyrille Flamant participated to the field experiment on board ATR-42 and contributed to the paper editing.

**Competing interests.** The authors declare that they have no conflict of interest.

**Acknowledgements.** The authors acknowledge Sandrine Bony who coordinated EUREC[4]A. The authors gratefully acknowledge Jean-Christophe Canonici, Jean-Christophe Desbios, Thierry Perrin, Laurent Guiraud and all the Technicians, Engineers, Pilots and Director from SAFIRE, the French facility for airborne research (http://www.safire.fr, last access: 17 December 2021), and Airplane Delivery, for making the preparation of the ATR and the EUREC[4]A airborne operations possible. Julien Totems is acknowledged for its help in preparing and participating in the field experiment. We thank the Caribbean Regional Security System (RSS) for hosting the ATR and the ATR team in Barbados during the experiment, Dr David Farrell and the Caribbean Institute for Meteorology and Hydrology (CIMH) for their logistical and administrative support, the Department of Civil Aviation in Barbados and Andrea Hausold (from DLR), for their help and support of airborne operations. The authors also thank AERIS for their support during the campaign and for managing the EUREC[4]A database. Maëlie Chazette is thanked for proofreading the article. The authors would like to thank the reviewers for their valuable comments which helped improving the quality of the manuscript.

**Financial support.** This research was supported by the European Research Council (ERC) under the European Union's Horizon 2020 research and innovation program (Grant Agreement No. 694768), with some additional support from by the French Space Agency CNES through the EECLAT project.

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
