# Peer review of "Mesoscale spatio-temporal variability of airborne lidarderived aerosol properties in the Barbados region during EUREC4A"

_Atmospheric Chemistry and Physics, 2021_

## Author Response (AR3)

**Response to Editor**

There is only one minor issue that I would like you to address before accepting your manuscript for publication: the description and interpretation of the CAMS modelling system could be improved. First, it is referred to as an "aerosol transport model" but CAMS goes beyond that and includes a sophisticated assimilation system. Hence, the finding that "The aerosol plume reproduced by the model is very close to the one actually observed by MODIS." is not very surprising as CAMS assimilates MODIS (with fairly small errors so at assimilation time it is very close to MODIS indeed). However, the well reproduced vertical structure is not based on explicit assimilation, which is something worth mentioning. As a whole, a short introduction to CAMS, including which relevant aerosol properties are assimilated and which are not would strengthen this aspect of the manuscript. This could also be pulled through into the relevant discussions.

Yes, the editor is right, the AOT MODIS data are assimilated into the CAMS model, but not the notion of vertical aerosol distribution. We have therefore clarified these points in the paper and have better highlighted the result as to the observed agreement between modelling and lidar observation for the vertical distribution of aerosols.

We have thus added the sentence " The aerosol vertical structures appear to be well reproduced using atmospheric composition reanalyses from CAMS when comparing with lidar-derived vertical profiles." In the abstract.

In subsection 4.2 we have nuanced our comments by modifying the text as follows: " As CAMS assimilate the MODIS-derived AOT (Benedetti et al., 2009; Inness et al., 2018), the aerosol plume reproduced by the model is very close to the one actually observed by MODIS. However, the advantage of the chemistry-transport model is that it provides the chemical and optical speciation of the aerosols in the plume."

We have also added the sentence "It is noteworthy that the vertical distribution of aerosols is not constrained in the assimilation process, as opposed to the horizontal distribution of aerosols." in subsection 4.4.

And we have changed the beginning of the conclusion: "This study has shown a very strong consistency between airborne lidar observations, passive and active satellite instrumentation. All these measurements are also in very good agreement with the CAMS aerosol reanalysis products, and in particular with regard to the vertical distribution of aerosols."

The authors would like to thank the reviewers for their valuable comments which helped improving the quality of the manuscript. Our point-by-point responses to the reviewer's comments appear in bold below. The text modified in the revised version of the MS and included in the response appears in red.

**Reviewer 1**

The results are in good agreement with many other studies performed within former field activities (SAMUM, SALTRACE), However, these previous activities are not mentioned. This should be improved. A paper is much more exciting when a more complex overview of forgoing work is given and how the presented work fits into the big picture and what are the new findings compared to the older ones.

**We agree this material is missing. We have included in the revised version of the MS most of the references suggested by the referee, and more, in connection with SAMUM-2, SALTRACE and NARVAL.**

Minor revisions are required and may further improve the paper.

Detailed comments and suggestions:

Abstract: The abstract should summarize observations and solid results!

Lines 21 to 25: Are these statements based on observations? ..... or is this just your conclusions (opinion) from your observations? I mean, I do not find the respective figures in which the strong spatial heterogeneity in the aerosol fields is clearly documented. Maybe, I overlooked it! I also do not find any (correlation) study in which the impact of relative humidity on the aerosol properties is presented.

**As the role of humidity is one hypothesis among others, we do not include it in the abstract. It is indeed too speculative. On the other hand, it is clear from the figures that the horizontal aerosol field represented by their optical properties is heterogeneous in the presence of aerosol plumes. In the background situation, this is indeed less true.**

Page 2, line 19: Please have a look into the SALTRACE overview article of Weinzierl et al., BAMS, 2017 (https://doi.org/10.1175/BAMS-D-15-00142.1), and check the many references regarding SAMUM 2 and SALTRACE, especially Haarig et al. (2017, ACP), Rittmeister et al. (2017, ACP), Tesche et al. (2011, Tellus) and (2009, JGR). There many more papers on smoke transport over the tropical Atlantic, in addition to the papers of Ansmann et al. and Baars et al. you mentioned already.

We have re-written the beginning of the introduction to take the suggested studies into account. To comply with similar comments made by Referee #2, we have included a discussion on the seasonal transport (and the difference between wintertime and summertime dust and BB transport). The part of the text related to the transport of smoke and the importance of dust-BB mixtures in the atmosphere composition is now introduced earlier in the Introduction, i.e. After the 1st sentence.

"Long-range transport of SD and BBA aerosols from West Africa across the equatorial North Atlantic occurs all year long, but exhibits a marked seasonal cycle. For instance, summertime and wintertime SD aerosol transport characteristics have been shown to differ significantly, with SD being transported at higher latitude and coarser particles being advected further west during the summer (van der Does et al., 2017) in the African Easterly jet-driven Saharan air layer (e.g. Prospero and Carlson, 1972). In contrast, during wintertime, SD is transported at lower altitudes (below 3 km amsl) and further south (owing to the equatorward migration of the intertropical Convergence Zone) towards northeast South America (e.g. Swap et al., 1992; Ansmann et al., 2009; Baars et al., 2011) and the Caribbean (Haarig et al., 2016). SD in the Caribbean is generally observed to be mixed with BB aerosols from West Africa and South America, with BB-SD mixtures generally being carried above dust layers in the winter (Tesche et al., 2009, 2011; Weinzeierl et al., 2017; Haarig et al., 2017).''

Page 3, line 3: After this paragraph we need a paragraph on all the SAMUM 2 and SALTRACE observations (maybe with focus on lidar only) of complex dust and smoke mixtures over the tropical Atlantic (from Africa to the Caribbean). Please have a look into the special issue of SAMUM 2 (in Tellus, 2011) and especially into the SALTRACE overview paper of Weinzierl et al. (2017). The results of the SALTRACE campaign plus the ship cruise (Rittmeister et al., 2017) must be considered later on in the discussion as well.

Agreed. We have significantly enhanced this part of the Introduction to include all relevant past field campaigns and references. This part of the Introduction now writes:

"These measurements were very soon followed by numerous lidar observations across the North Atlantic acquired as part of dedicated campaigns such as SAMUM-2 (Saharan Mineral Dust Experiment, Ansmann et al., 2011), SALTRACE (Saharan Aerosol Long-Range Transport and Aerosol–Cloud-Interaction Experiment, Weinzierl et al., 2017) and NARVAL (Next-generation Aircraft Remote-Sensing for Validation Studies, Stevens et al., 2019). Such observations were made from ground-based lidar measurements in the Cape Verde region (Ansmann et al., 2009, 2011), in Barbados (Groß et al., 2015; Haarig et al., 2017) and over Amazonia (e.g. Ansmann et al., 2009; Baars et al., 2011), from shipborne lidar measurements (Rittmeister et al., 2017) and from nadir-pointing airborne lidar measurements (Chazette et al., 2001; Tanré et al., 2003; Weinzierl et al., 2011, 2017; Gutleben et al., 2019)."

Page 5, line 5: Please provide some information about the regression function V0. What do you exactly mean with this regression function?

**This term is not appropriate, we have replaced by "linear fitting".**

Page 5, lines 20-24: Tesche et al. (2009, 2011) already studied complex mixtures of dust and smoke during the high winter months (January-February), but over Cabo Verde in 2008. Haarig et al. used the Caribbean SALTRACE winter campaign in Februray-March 2014 to study again dust-smoke mixtures coming from Africa.

These references are now included in the Introduction. The immediate link with the MS text cited here is not clear to us, as on page 5 lines 20-24 we only introduce the type of aerosols composition context in which the ATR42 aircraft flew during EUREC4A. They are also added later on in Section 4.3 when the CALIOP-derived aerosol identification is discussed.

A new, at least not well studied aspect you may want to stress in more detail is the following: Usually it is assumed that dust is transported towards South America in January- February, south of the Caribbean, and the tropical Atlantic towards North America is dust free, but you show that this is not (or no longer) the case. Big plumes of dust and smoke (because of the dry season or burning season in central western Africa) are transported even towards North America during wintertime.

**Indeed, we find a more northerly transport than usually reported in the scientific literature. We have emphasized this point in the abstract, in section 5 and in the conclusion.**

In the figures captions (or maybe in the plots), one should provide dates and also the times of observations (periods in UTC).

**The missing date and time of the flights have been added in the figure captions.**

'Terrigenous' is a bit unspecific, you mean: dust? Or even smoke from continents? You want to say, non-marine aerosol?

**Agree, terrigenous has been replaced by dust.**

I appreciate the exhausting analysis, including all the MODIS, CALIOP, and CAMS products!

**Thank you.**

Page 26: In the discussion section 5, one should integrate the SAMUM and SALTRACE findings, what was similar, what are the news points of your study (additional and complementary aspects). Such a discussion will improve significantly the visibility of this paper later on.

This is now done. In particular, we are now comparing our findings related to aerosol optical properties to those of Haarig et al. (2017, 2019) in Section 3 also. It is worth noting that there are very few studies of the aerosol optical properties in Barbados in wintertime. Hence, we agree that is important to mention them.

All in all, a good study based on high quality observations, analysed by experienced scientists!

**Thanks again for your constructive comments.**

**References added in the text:**

Ansmann, A., Petzold, A., Kandler, K., Tegen, I., Wendisch, M., Müller, D., Weinzierl, B., Müller, T., and Heintzenberg, J.: Saharan Mineral Dust Experiments SAMUM–1 and SAMUM–2: What have we learned?, Tellus B, 63, 403–429, 2011.

Groß, S., Freudenthaler, V., Schepanski, K., Toledano, C., Schäfler, A., Ansmann, A., and Weinzierl, B.: Optical properties of long-range transported Saharan dust over Barbados as measured by dualwavelength depolarization Raman lidar measurements, Atmos. Chem. Phys., 15, 11067–11080, https://doi.org/10.5194/acp-15-11067-2015, 2015. Gutleben, M., Groß, S., and Wirth, M.: Cloud macro-physical properties in Saharan-dust-laden and dust-free North Atlantic trade wind regimes: a lidar case study, Atmos. Chem. Phys., 19, 10659–10673, https://doi.org/10.5194/acp-19-10659-2019, 2019.

Haarig, M., Ansmann, A., Gasteiger, J., Kandler, K., Althausen, D., Baars, H., Radenz, M., and Farrell, D. A.: Dry versus wet marine particle optical properties: RH dependence of depolarization ratio, backscatter, and extinction from multiwavelength lidar measurements during SALTRACE, Atmos. Chem. Phys., 17, 14199–14217, https://doi.org/10.5194/acp-17-14199-2017, 2017.

Haarig, M., Walser, A., Ansmann, A., Dollner, M., Althausen, D., Sauer, D., Farrell, D., and Weinzierl, B.: Profiles of cloud condensation nuclei, dust mass concentration, and ice-nucleatingparticle-relevant aerosol properties in the Saharan Air Layer over Barbados from polarization lidar and airborne in situ measurements, Atmos. Chem. Phys., 19, 13773–13788, https://doi.org/10.5194/acp-19-13773-2019, 2019.

Kim, S.-W., Yoon, S.-C., Chung, E.-S., Sohn, B.-J., Berthier, S., Raut, J.-C., Chazette, P. and Dulacb, F.: Initial assessment of space-based lidar CALIOP aerosol and cloud layer structures through intercomparison with a ground-based back-scattering lidar and CloudSat, in AIP Conference Proceedings, vol. 1100., 2009.

Prospero, J. and T. N. Carlson, 1972: Vertical and areal distribution of Saharan dust over western equatorial north Atlantic Ocean. J. Geophys. Res., 77, 5255–5265, doi:10.1029/JC077i027p05255.

Rittmeister, F., Ansmann, A., Engelmann, R., Skupin, A., Baars, H., Kanitz, T., and Kinne, S.: Profiling of Saharan dust from the Caribbean to western Africa – Part 1: Layering structures and optical properties from shipborne polarization/Raman lidar observations, Atmos. Chem. Phys., 17, 12963–12983, https://doi.org/10.5194/acp-17-12963-2017, 2017.

Stevens, B., Ament, F., Bony, S., Crewell, S., Ewald, F., Groß, S., Hansen, A., Hirsch, L., Jacob, M.,
Kölling, T., Konow, H., Mayer, B., Wendisch, M., Wirth, M., Wolf, K., Bakan, S., BauerPfundstein,
M., Brueck, M., Delanoë, J., Ehrlich, A., Farrell, D., Forde, M., Gödde, F., Grob, H., Hagen, M., Jäkel,
E., Jansen, F., Klepp, C., Klingebiel, M., Mech, M., Peters, G., Rapp, M., Wing, A. A., and Zinner, T.: A
high-altitude long-range aircraft configured as a cloud observatory – the NARVAL expeditions, B.
Am. Meteorol. Soc., 100, 1061–1077, https://doi.org/10.1175/bamsd-18-0198.1, 2019.

Swap, R., M. Garstang, S. Greco, R. Talbot, and P. Kallberg, 1992: Saharan dust in the Amazon basin. Tellus, 44B, 133–149, doi:10.1034/j.1600-0889.1992. t01-1-00005.x.

Tesche, M., Ansmann, A., Müller, D., Althausen, D., Engelmann, R., Freudenthaler, V., and Groß, S.: Vertically resolved separation of dust and smoke over Cape Verde using multiwavelength Raman and polarization lidars during Saharan Mineral Dust Experiment 2008, J. Geophys. Res., 114, D13202, https://doi.org/10.1029/2009JD011862, 2009

Tesche, M., Gross, S., Ansmann, A., Müller, D., Althausen, D., Freudenthaler, V., and Esselborn, M.: Profiling of Saharan dust and biomass-burning smoke with multiwavelength polarization Raman lidar at Cape Verde, Tellus B, 63, 649–676, https://doi.org/10.1111/j.1600-0889.2011.00548.x, 2011.

van der Does, M., Korte, L. F., Munday, C. I., Brummer, G.-J. A., and Stuut, J.-B. W.: Particle size traces modern Saharan dust transport and deposition across the equatorial North Atlantic, Atmos. Chem. Phys., 16, 13697–13710, https://doi.org/10.5194/acp-16-13697-2016, 2016.

Weinzierl, B., Ansmann, A., Prospero, J. M., Althausen, D., Benker, N., Chouza, F., Dollner, M., Farrell, D., Fomba, W. K., Freudenthaler, V., Gasteiger, J., ß, S. G., Haarig, M., Heinold, B., Kandler, K., Kristensen, T. B., Mayol-Bracero, O. L., Müller, T., Reitebuch, O., Sauer, D., Schäfler, A., Schepanski, K., Spanu, A., Tegen, I., Toledano, C., and Walser, A.: The Saharan Aerosol Long-range Transport and Aerosol-Cloud Interaction Experiment (SALTRACE): overview and selected highlights, B. Am. Meteorol. Soc., 98, 1427–1451, https://doi.org/10.1175/BAMS-D-15-00142.1, 2017. The authors would like to thank the reviewers for their valuable comments which helped improving the quality of the manuscript. Our point-by-point responses to the reviewer's comments appear in bold below. The text modified in the revised version of the MS and included in the response appears in quotes.

**Reviewer 2**

The manuscript is well in the focus of ACP and should be published after mainly minor revisions.

**General comments:**

In the introduction and the discussion findings of the approx. last 10 years on Saharan dust are completely missing. Large research projects and campaigns were conducted focusing on the beginning of dust transport (e.g. SAMUM – which focuses also on wintertime conditions), as well as after long-range transport towards the Caribbean (e.g. SALTRACE, NARVAL-II). A large number of studies were published using data from these studies that should and could be connected to the findings of this manuscript. Those studies were also dealing with wintertime dust transport, mixtures of dust and biomass burning aerosols, downward mixing of dust, and on the relation of Saharan dust layers and relative humidity. Those studies should be mentioned in the introduction and discussed in relation to the findings described in this manuscript.

We agree. This is now addressed in the revised version of the MS. We have re-written the beginning of the introduction to take the suggested studies into account with a discussion on the seasonal transport (and the difference between wintertime and summertime dust and BB transport). The part of the text related to the transport of smoke and the importance of dust-BB mixtures in the atmosphere composition is now introduced earlier in the Introduction, i.e. after the 1st sentence. To comply with similar comments made by Referee #1, we have included a number of references pertaining to SAMUM, SALTRACE and NARVAL.

**Specific comments:**

In the abstract information on what characterized the two distinct periods with significant aerosol content should be given. How is the heterogeneity connected to the highly variable relative humidity field?

**The second part of the abstract has been modified to take into account the referee's comment. As the role of humidity is one hypothesis among others, we do not include it in the abstract. It is indeed too speculative.**

Introduction: A differentiation between summertime and wintertime transport should be made. The main dust transport towards the Caribbean is happening in summertime (which is also mentioned in the manuscript). The Saharan Air Layer seems to be quite undisturbed close to the source and during long-range transport during the summertime transport. A number of publications (e.g. Weinzierl; Haarig; Groß; Gutleben; ...) described the summertime dust transport to the Caribbean. In contrast, during wintertime the dust is located at lower altitudes and frequently mixed with biomass burning aerosols (e.g. SAMUM-II related publications: Ansmann; Tesche; Groß; ...). Additionally, biomass burning might be transported to the Caribbean from the South American continent (Haarig).

**Agreed. This is now accounted for in the Introduction as mentioned above.**

Calibration of the lidar signal: How stable is the system constant when pressure and temperature change during flights? How do system settings affect the system constant?

We did not note any variation in the lidar calibration constant as a function of pressure for flights below 5 km a.m.s.l., as the cabin is pressurized. The temperature in the cabin was also stable during the flights. These two aspects have been added. As explained, the most important factor affecting the system constant is the transmission of the aircraft window, the variability of which could be assessed from the reflection of the laser beam from its surface.

Observation periods: The detailed information on the different observation periods should be given together to get a better overview of the different aerosol situations. The different observation periods should be described in a bit more detail. Which aerosols / mixtures were the dominant one? Or why were these periods chosen for a detailed description?

The selected periods are presented at the beginning of Section 3 and representatively sample the whole airborne campaign in terms of aerosol loads. At this stage, the reasons for this choice are roughly explained. The choice is confirmed as the observations and modelling are presented. A strong confirmation of this choice is given in subsection 4.4 by the CAMS modelling which is shown to be consistent with the spaceborne observations in subsection 4.2. We have added the sentence "It is worth noting that the selected observation periods are representative of the main aerosol situations, with rather contrasting relative contributions of aerosol compositions" in subsection 4.4 to emphasize the representativeness of the selected flights.

Is the vertical profile derived from the ascends and descends? Can you give a bit more information?

**Profiles include both. We have added the sentence "It should be noted that the vertical profiles include the ascent and descent parts of the flights."**

Page 6, line 31: Do you mean vertically homogenously distributed? I do not see it for horizontally...

**Yes, it is for the vertical. The correction has been done.**

Page 7, lines 8: How do you link the horizontal relative humidity filed to the particle's horizontal heterogeneity? Might it also be the other way around? As described in Gutleben et al., 2020, Saharan dust transport is associated with transport of embedded water vapor. To link relative humidity and aerosol heterogeneity one needs to have information about the water vapor / relative humidity field, and on the type of particle. To better describe the vertical distribution and the connection to possible convective processes a consideration of the atmospheric stability would be helpful (e.g. inversions, stability).

**Yes, more information is needed to properly study the links between water vapour and the optical properties of aerosols. In this part of the text, we only state the causes that**

**could explain the observed heterogeneity and humidity is one of them, as well as sources diversity and vertical mixing by convection.**

Page 11, lines 14: To connect changes in relative humidity to changes in the optical properties, information on the relative humidity distribution is needed. Furthermore, e.g. dust aerosols are not hydrophil. Thus, relative humidity should not affect the intensive optical properties. What about biomass burning aerosols? What kind of mixtures do you consider?

As before, it is hypothesised that RH may influence the variability of LR and PDR. It is also said that LR varies little with RH and that the observed variability may also be related to different aerosol natures. This does not exclude an effect of RH, but it is difficult to quantify at this level. We have added an explanation in subsection 4.2 where the 2 main aerosol components are presented via CAMS modelling:

"The simultaneous presence of dusts and biomass burning aerosols may explain the heterogeneous character observed above. This does not exclude a role of relative humidity as explained by Kim et al. (2009) for the winter period in West Africa. They have shown that the biomass burning aerosol plumes advected over long distances are associated with significantly higher relative humidity values than the dust plumes. These two plumes may co-exist at different altitudes or be mixed as in our case. This mixture may not be homogeneous."

It is true that it is often said that dusts are hydrophobic, but this is not necessarily the case in a mixture of aerosols, as in our case, where nitrates can be positioned on the aerosol surface. For this study, we cannot discuss this aspect because no aerosol chemistry measurements were performed.

Figure 5: Capture is missing the date information.

Yes, the date information has been added in the figure caption.